# Mineralogy and Geochemistry of the Upper Ordovician and Lower Silurian Wufeng-Longmaxi Shale on the Yangtze Platform, South China: Implications for Provenance Analysis and Shale Gas Sweet-Spot Interval

Zhensheng Shi [1] , Shengxian Zhao [2], Tianqi Zhou [1,*], Lihua Ding [1], Shasha Sun [1] and Feng Cheng [1]

1   Petro China Research Institute of Petroleum Exploration and Development, Beijing 100083, China
2   Shale Gas Research Institution, Southwest Oli & Gas Field Company, PetroChina, Chengdu 610051, China
*   Correspondence: zhoutianqi@petrochina.com.cn

**Abstract:** The sediment provenance influences the formation of the shale gas sweet-spot interval of the Upper Ordovician–Lower Silurian Wufeng–Longmaxi shale from the Yangtze Platform, South China. To identify the provenance, the mineralogy and geochemistry of the shale were investigated. The methods included optical microscopy analysis, X-ray diffraction testing, field-emission scanning electron imaging, and major and trace element analysis. The Wufeng–Longmaxi shale is mainly composed of quartz (avg. 39.94%), calcite (avg. 12.29%), dolomite (avg. 11.75%), and clay minerals (avg. 28.31%). The LM1 interval is the shale gas sweet-spot and has the highest contents of total quartz (avg. 62.1%, among which microcrystalline quartz accounts for 52.8% on average) and total organic carbon (avg. 4.6%). The relatively narrow range of $TiO_2$–Zr variation and the close correlation between Th/Sc and Zr/Sc signify no obvious sorting and recycling of the sediment source rocks. Sedimentary sorting has a limited impact on the geochemical features of the shale. The relatively high value of ICV (index of compositional variability) (1.03–3.86) and the low value of CIA (chemical index of alteration values) (50.62–74.48) indicate immature sediment source rocks, probably undergoing weak to moderate chemical weathering. All samples have patterns of moderately enriched light rare-earth elements and flat heavy rare-earth elements with negative Eu anomalies (Eu/Eu* = 0.35–0.92) in chondrite-normalized diagrams. According to Th/Sc, Zr/Sc, La/Th, $Zr/Al_2O_3$, $TiO_2$/Zr, Co/Th, $SiO_2/Al_2O_3$, $K_2O/Na_2O$, and La/Sc, it can be inferred that the major sediment source rocks were acidic igneous rocks derived from the active continental margin and continental island arc. A limited terrigenous supply caused by the inactive tectonic setting is an alternative interpretation of the formation of the sweet-spot interval.

**Keywords:** provenance analysis; geochemistry; Wufeng Formation; Longmaxi Formation; Yangtze Platform

## 1. Introduction

A series of fine-grained sediments, such as mudstone, shale, bituminous shale, carbonaceous shale, and even carbonaceous diatomite, can be classified as shales [1–4]. Late Ordovician and Early Silurian black shales, which form an important shale interval after the Phanerozoic, are distributed on continental shelves worldwide [5–13]. Several geological events, such as Oceanic Anoxic Events (OAEs) and the Late Ordovician Mass Extinction (LOME), have been recognized in this interval, e.g., [10,11,14–16]. The major force behind the events was an abrupt rise in temperature caused by a rapid increase in carbon dioxide content in the atmosphere because of volcanogenic/methanogenic activity [11,17–21]. Rapid global warming accelerated land weathering and nutrient input to the ocean, triggering ocean eutrophication and global anoxia/euxinia [11]. The mineralogy and geochemistry of shale, which is significantly influenced by its provenance, can be used

to indicate the weathering, paleoclimate, and volcanogenic/methanogenic activity of the sediment source area [14].

The Wufeng–Longmaxi shale on the Yangtze Platform is an important part of the Upper Ordovician and Lower Silurian black shales. The shale is heterogeneous in chemical composition and is characterized by relatively high percentages of quartz, carbonate minerals, and organic matter, as well as by the significant enrichment of several trace elements (e.g., Ni, Zn, U, and Ba) and rare-earth elements (REEs), e.g., [22,23]. As the most successful shale gas play in China and one of the largest shale gas plays worldwide [24–26], the production from shale reached over $229 \times 10^9$ m$^3$ in 2021 [27]. Although the thickness of the shale is generally more than 300 m, the shale gas is predominantly produced from the organic-rich interval at the bottom of this shale, with a thickness of 10–40 m [28] or even just a few meters [29]. The gas-producing interval is commonly referred to as the "sweet-spot" interval in unconventional petroleum exploration and development [28]. The sweet-spot interval is characterized by high total organic carbon (TOC) and gas contents, relatively high porosity, high brittle mineral content, and abundant lamination fissures and fractures [28,29]. The formation of the interval was possibly controlled by an anoxic shelf environment [15,23], sealed roof and floor strata [28], and abundant lamination [29]. However, the effect of provenance on the formation of the shale gas "sweet-spot" interval is rarely mentioned.

Based on the investigation of mineralogy and geochemistry, this paper aims to evaluate potential variations due to weathering, sorting, and recycling to constrain the provenance and tectonic setting of the Wufeng–Longmaxi shale on the Yangtze Platform. The findings from this study can provide a reasonable explanation for the formation of the sweet-spot interval of the Wufeng–Longmaxi shale, as well as other shales of the same age worldwide, e.g., [6].

## 2. Geological Setting

The Upper Ordovician and Lower Silurian black shales on the Yangtze Platform were deposited during the demise of the South China Basin and the formation of the South China orogenic belt [30–32]. At the end of the Cambrian, the Cathaysian and the Yangtze blocks converged because of the Guangxi orogeny, and the Southeast Yangtze Platform and Jiangnan Basin were raised in succession [32–34]. Then, the Yangtze region recorded the development history of the passive continental margin [32]. A carbonate platform was the main sedimentary facies in the Yangtze region during the Early to Middle Ordovician [35,36]. Beginning with the Late Ordovician, a mixed carbonate–clastic epeiric sea covered the Yangtze basin, which was commonly scattered along the intrashelf sub-basin. A paleogeographic reconstruction showed significant sea-level changes during the Ordovician–Silurian transition [37]. In the Early Silurian, the Cathaysia plate began to expand, and most of South China rose to the land, resulting in great changes in the distributions of land and sea in South China (Figure 1). The Yangtze Sea became semi-closed due to the rise of ancient lands, and a subaqueous high was formed under the action of regional tectonic stress [37]. Due to the barrier of the land, uplifts, and the high topography of the seafloor, the Yangtze Sea evolved into a deep-water shelf facies with an anoxic and stagnated water column [38,39].

The Wufeng–Longmaxi shale on the Yangtze Platform is widely distributed and has a total thickness reaching up to 500 m [24,40]. The shale is divided into the Wufeng Formation, the Guanyinqiao Bed, and the Longmaxi Formation (Figure 2). The Wufeng Formation is parallel unconformable with the underlying Baota nodular limestone and parallel conformable with the overlying Guanyinqiao Bed (Figure 2) [26]. The Longmaxi Formation, which is conformable with the Guanyinqiao Bed, is divided into Member 1 and Member 2, and Member 1 is subdivided into Sub-member 1 and Sub-member 2 [26]. Sub-member 1, which is characterized by high contents of organic matter, laminae, and microfractures [29,41], is the sweet-spot interval for shale gas exploration and develop-

ment [42]. In addition, the Wufeng–Longmaxi shale can be divided into thirteen graptolite biozones, two 2nd cycles, and four 3rd cycles [24].

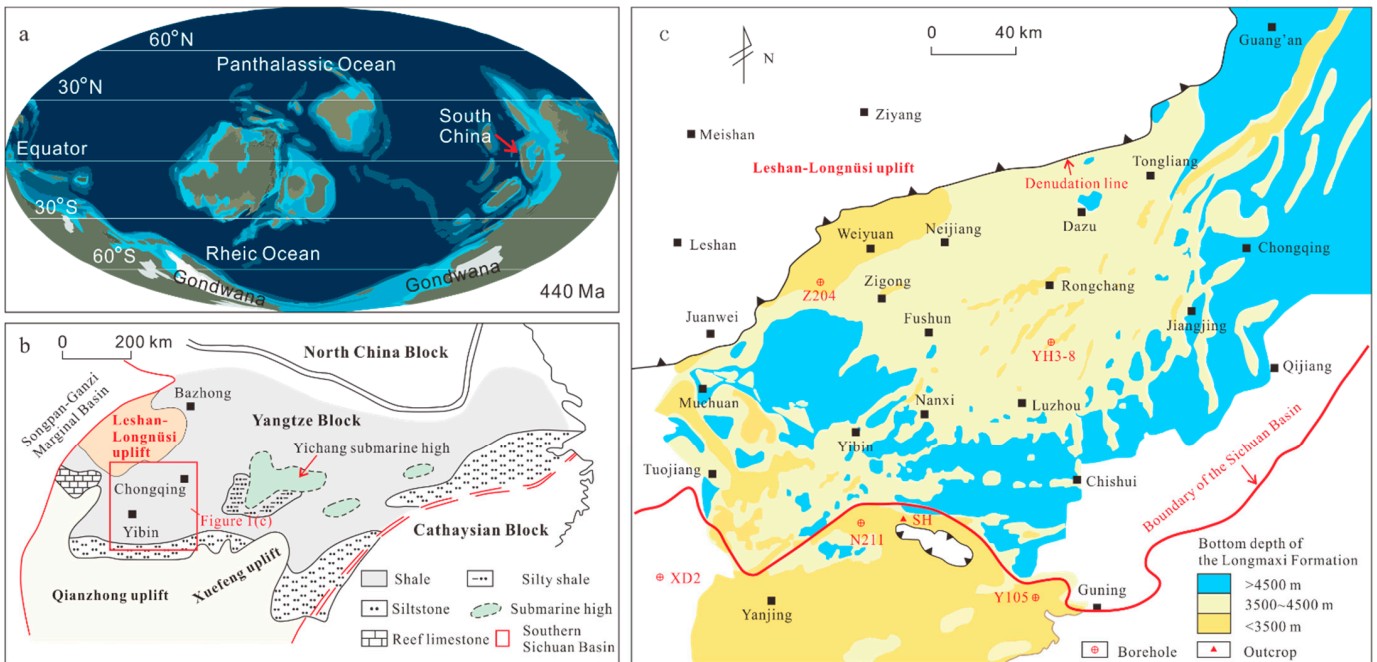

**Figure 1.** (**a**) Global paleogeography during the Early Silurian (the map originates from https://deeptimemaps.com (accessed on 18 July 2022)); (**b**) paleogeographic map of the Yangtze shelf sea during the Early Silurian showing the location of the southern Sichuan Basin (modified from [37]); (**c**) the bottom depth of the Longmaxi Formation of the southern Sichuan Basin (modified from [43]) (For interpretation of the references used to color in this figure legend, the reader is referred to the Web version of this article).

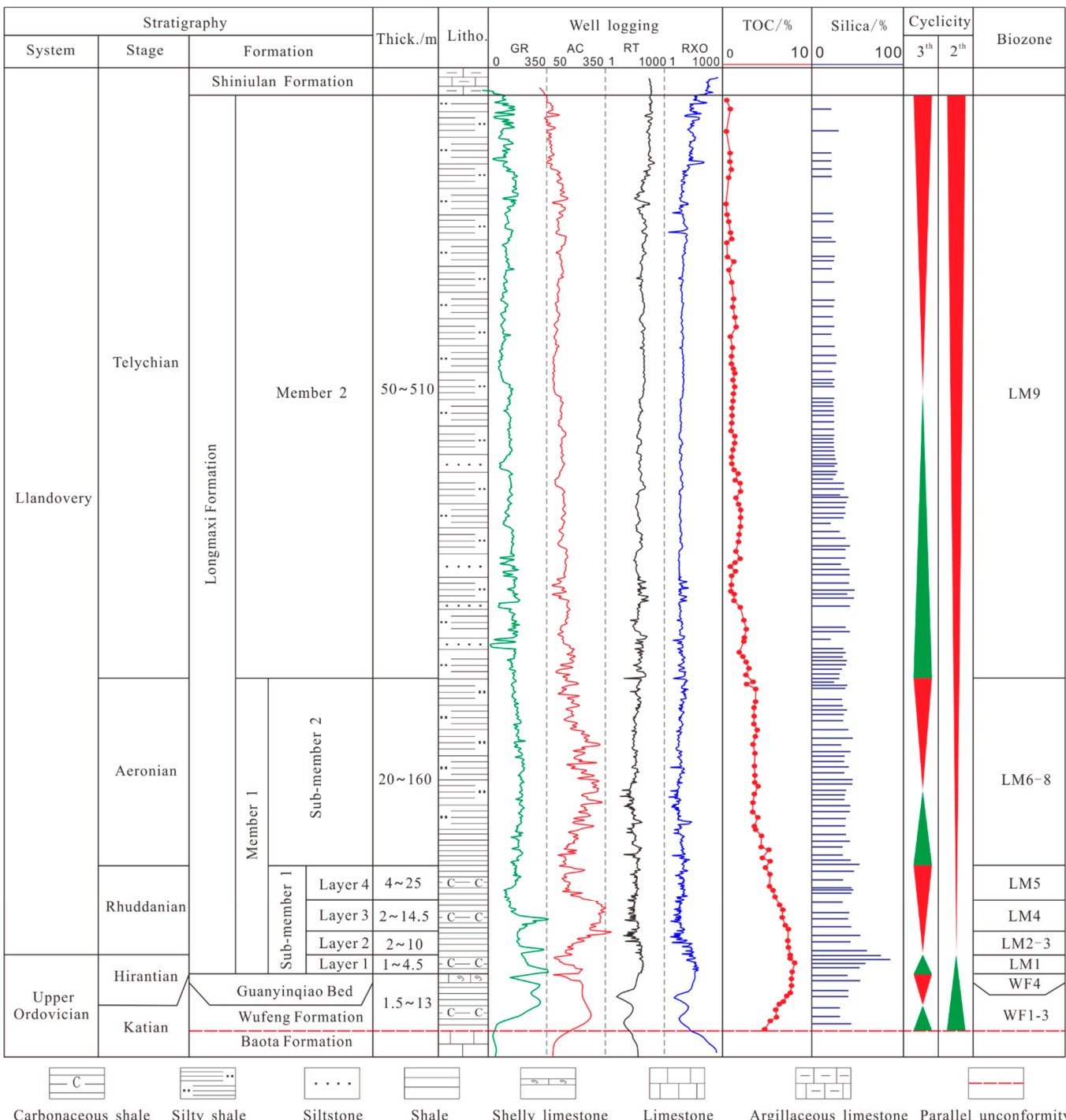

**Figure 2.** Stratigraphic column and study interval of the Wufeng–Longmaxi shale (modified from [24]). WF1 refers to the *Dicellograptus complanatus* biozone, WF2 refers to the *Dicellograptus complexus* biozone, WF3 refers to the *Paraorthograptus pacificus* biozone, and WF4 refers to the *Metabolograptus extraordinarius* biozone; LM1 refers to the *Metabolograptus persculptus* biozone, LM2 refers to the *Akidograptus ascensus* biozone, LM3 refers to the *Parakidograptus acuminatus* biozone, LM4 refers to the *Cystograptus vesiculosus* biozone, LM5 refers to the *Coronograptuscyphus* biozone, LM6 refers to *Demirastrites triangulates*, LM7 refers to the *Lituigraptus convolutes* biozone, LM8 refers to the *Stimulograptus sedgwickii* biozone, and LM9 refers to the *Spirograptus guerichi* biozone [44].

## 3. Samples and Methods

In this study, a total of eighty-three black shale samples were collected from wells Y105, XD2, Z204, YH3-8, and N211 and the SH outcrop (Figure 1c). All samples are organic-rich black shales (Figure 3) collected from the Wufeng Formation, the Guangyinqiao bed, and the lowermost part of the Longmaxi Formation.

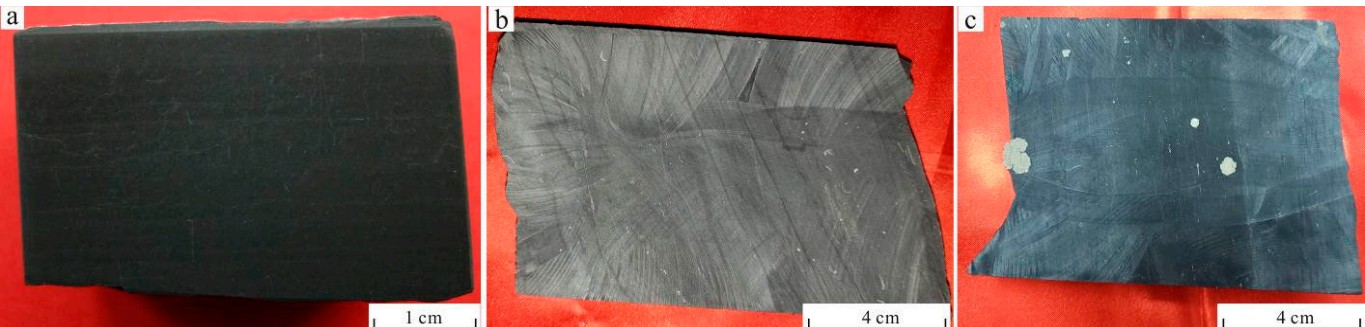

**Figure 3.** Core photographs from (**a**) SH outcrop (sample number: SH5-26-2), (**b**) well YH3-8 (sample number: YH-236), and (**c**) well Z204 (sample number: Z4-8) showing characteristics of the Wufeng-Longmaxi shale (For the locations of the wells and outcrop, the reader is referred to Figure 1 of this article).

### 3.1. Mineral Composition and Morphological Analysis

X-ray diffraction (XRD) was performed on 83 shale sample powders. XRD measurement was conducted with a Panalytical X–Pert PRO MPD X-ray diffractometer from Amsterdam, Netherlands, at a working voltage of 50 keV and a current of 800 μA. Diffractograms were recorded from 5° to 90° at a rate of 2θ. Sample preparation and spectral identification followed the Chinese oil and gas industry standard (SY/T) 5163-2014. After that, the mineral contents of quartz, calcite, dolomite, clay minerals, feldspar, and pyrite were determined.

Field-emission scanning electron microscopy (FE–SEM) in conjunction with an energy-dispersive X-ray spectrometer (EDAX, New York, NY, USA) was utilized to study the mineral morphology and contents. Twenty-five shale samples were mechanically polished and then polished with Ar ions on a 600 DuoMill instrument (Gatan, New York, NY, USA) at 4 KV and a low angle (7.5°) for 2 h. Details about milling and SEM observations can be found in previous studies [29].

### 3.2. Geochemistry Analysis

For total organic carbon (TOC) content measurement, 83 shale samples were decarbonized by soaking them in 4 M HCl at 60 °C for at least 24 h. After that, impurities and HCl were removed by rinsing in distilled water and then dried. TOC content was determined by a LECO CS-400 analyzer (LECO, New York, NY, USA), and the standard deviation of the measurements was lower than ±0.10%.

Major element concentrations were measured by X-ray fluorescence (XRF). Eighty-three shale samples were ground to 200 mesh in an agate mortar, and 1.2 g of each sample was accurately weighed after drying in a drying box. Then, 6 g of solvent ($Li_2B_4O_7$) was added and fully mixed in a milk bowl, and the mixture was moved into a platinum crucible and dissolved into a uniform glass sheet at a high temperature of 1100 °C. Then, the major components of the glass sheet were tested using an XRF-1500 spectrometer (ThermoFisher, New York, NY, USA).

Trace element concentrations were analyzed by inductively coupled plasma mass spectrometry (ICP–MS, AMETEK, Berlin, Germany). Samples (100 mg) were dried at 105 °C and then digested with a reagent composed of 0.5 mL of $HClO_4$, 2.5 mL of HF, and 0.5 mL of $HNO_3$ as well as 1 mL of $HNO_3$ and 3 mL of $H_2O$. After that, the solution was diluted and measured by ICP–MS. A replicate analysis of the samples indicated better than

2% precision for the analysis of major oxides and 4% for the elements analyzed by ICP–MS. The standard samples OU–6 (slate), AMH–1 (andesite), and GBPG–1 (plagiogneiss) were utilized to monitor the analysis. The analytical precision of the trace elements was better than 0.5%.

Major and trace element results were contrasted with the post–Archean Australian shales (PAAS, [45]) and Phanerozoic North American shale composite (NASC) [46].

### 3.3. Depositional Structures Analysis

Twenty-five pieces of thin sections for macroscopic depositional structure analysis were prepared with sizes of 7 cm × 5 cm. The thickness of the thin sections was about 15–20 microns, and the maximum thickness was less than 30 microns. Depositional structures were mainly described by full-size thin-section imaging and polarized light microscopy. Details about full-size large thin-section imaging and polarized light microscopy can be found in previous studies [29,47].

## 4. Results
### 4.1. Mineral Compositions

The Wufeng-Longmaxi shale is mainly composed of quartz, carbonate minerals, and clay minerals, with minor amounts of feldspar and pyrite (Table 1). In addition, trace amounts of apatite and barite can be observed in some samples.

The content of quartz ranges from 22.0% to 73.0%, with an average of 39.94%. There exist silt-sized quartz and clay-sized quartz. The silt-sized quartz is commonly composed of fine silt grains (grain sizes between 3.9 and 31.2 μm) with angular or sub-angular edges (Figure 4a). The clay-sized quartz includes microcrystalline quartz and quartz associated with clay minerals. The microcrystalline quartz has a grain size below 4 μm (Figure 4b). The quartz associated with clay minerals is commonly irregularly shaped and coexists with clay minerals (Figure 4c). The LM1 interval has the highest content of quartz (avg. 62.1%; Table 1), in which microcrystalline quartz can reach up to 85% (avg. 52.8%). For the Longmaxi Formation, the content of clay-sized quartz decreases, and that of silt-sized quartz increases progressively upwards (Figure 5).

Table 1. Mineral compositions and TOC contents of the Wufeng–Longmaxi shale on the Yangtze Platform, South China.

| No. | Well | Depth | Lithology | Formation | Graptolite Zone | Quartz | Calcite | Dolomite | Clay Minerals | Potassium Feldspar | Plagioclase | Pyrite | TOC |
|-----|------|-------|-----------|-----------|-----------------|--------|---------|----------|---------------|--------------------|-------------|--------|-----|
| 1 | Y105 | 1650.5 | Shale | Longmaxi | LM6 | 39.0 | / | / | 45.0 | / | 8.0 | 2.0 | 0.5 |
| 2 | Y105 | 1655.1 | Shale | Longmaxi | LM6 | 39.0 | / | 1.0 | 50.0 | / | 7.0 | 2.0 | 0.6 |
| 3 | Y105 | 1658.4 | Shale | Longmaxi | LM6 | 36.0 | / | 1.0 | 51.0 | / | 7.0 | 3.0 | 0.8 |
| 4 | YH3–8 | 3742.4 | Shale | Longmaxi | LM6 | 38.0 | / | / | 53.0 | / | 7.0 | 1.0 | 0.3 |
| 5 | YH3–8 | 3744.4 | Shale | Longmaxi | LM6 | 36.0 | 1.0 | 4.0 | 54.0 | / | 7.0 | 2.0 | 0.80 |
| 6 | YH3–8 | 3746.9 | Shale | Longmaxi | LM6 | 43.0 | / | / | 50.0 | / | 8.0 | 3.0 | 0.66 |
| 7 | YH3–8 | 3747.3 | Shale | Longmaxi | LM6 | 36.0 | / | 3.0 | 46.0 | / | 8.0 | 3.0 | 0.70 |
| 8 | XD2 | 2052.5 | Shale | Longmaxi | LM5 | 40.0 | / | 0.0 | 50.0 | / | 8.0 | 3.0 | 0.77 |
| 9 | XD2 | 2055.3 | Shale | Longmaxi | LM5 | 36.0 | 2.0 | 3.0 | 49.0 | / | 8.0 | 2.0 | 1.09 |
| 10 | Y105 | 1664.1 | Shale | Longmaxi | LM5 | 34.0 | 2.0 | 3.0 | 49.0 | / | 10.0 | 2.0 | 0.83 |
| 11 | Y105 | 1668.1 | Shale | Longmaxi | LM5 | 39.0 | 0.0 | / | 49.0 | / | 7.0 | 3.0 | 0.63 |
| 12 | Y105 | 1673.1 | Shale | Longmaxi | LM5 | 36.0 | 2.0 | 1.0 | 51.0 | / | 6.0 | 2.0 | 1.00 |
| 13 | YH3–8 | 3749.4 | Shale | Longmaxi | LM5 | 34.0 | 3.0 | 12.0 | 53.0 | / | 6.0 | 2.0 | 0.95 |
| 14 | YH3–8 | 3753.4 | Shale | Longmaxi | LM5 | 29.0 | 3.0 | 11.0 | 43.0 | / | 7.0 | 3.0 | 1.22 |
| 15 | YH3–8 | 3757.1 | Shale | Longmaxi | LM5 | 35.0 | 3.0 | 5.0 | 47.0 | / | 7.0 | 3.0 | 1.38 |
| 16 | YH3–8 | 3761.7 | Shale | Longmaxi | LM5 | 37.0 | 2.0 | 4.0 | 47.0 | / | 6.0 | 5.0 | 2.06 |
| 17 | YH3–8 | 3766.0 | Shale | Longmaxi | LM5 | 26.0 | 12.0 | 13.0 | 46.0 | / | 8.0 | 3.0 | 2.54 |
| 18 | Z204 | 3398.6 | Shale | Longmaxi | LM4 | 33.0 | 4.0 | 6.0 | 38.0 | / | 8.0 | 2.0 | 1.53 |
| 19 | XD2 | 2058.6 | Shale | Longmaxi | LM4 | 23.0 | 15.0 | 33.0 | 44.0 | / | 9.0 | 2.0 | 2.70 |
| 20 | XD2 | 2061.1 | Shale | Longmaxi | LM4 | 34.0 | 11.0 | 12.0 | 18.0 | / | 8.0 | 2.0 | 2.83 |
| 21 | XD2 | 2062.6 | Shale | Longmaxi | LM4 | 34.0 | 10.0 | 9.0 | 33.0 | / | 8.0 | 3.0 | 2.04 |
| 22 | Y105 | 1674.6 | Shale | Longmaxi | LM4 | 27.0 | 16.0 | 12.0 | 36.0 | / | 7.0 | 3.0 | 2.06 |
| 23 | Y105 | 1676.3 | Shale | Longmaxi | LM4 | 34.0 | 8.0 | 16.0 | 35.0 | / | 7.0 | 2.0 | 2.39 |
| 24 | Y105 | 1677.7 | Shale | Longmaxi | LM4 | 28.0 | 10.0 | 5.0 | 33.0 | / | 7.0 | 8.0 | 2.90 |
| 25 | Y105 | 1679.5 | Shale | Longmaxi | LM4 | 24.0 | 14.0 | 9.0 | 42.0 | / | 7.0 | 4.0 | 2.13 |
| 26 | Y105 | 1681.8 | Shale | Longmaxi | LM4 | 25.0 | 16.0 | 12.0 | 42.0 | / | 4.0 | 4.0 | 2.04 |
| 27 | YH3–8 | 3769.8 | Shale | Longmaxi | LM4 | 30.0 | 3.0 | 2.0 | 39.0 | / | 8.0 | 6.0 | 2.78 |

**Table 1.** *Cont.*

| No. | Well | Depth | Lithology | Formation | Graptolite Zone | Quartz | Calcite | Dolomite | Clay Minerals | Potassium Feldspar | Plagioclase | Pyrite | TOC |
|---|---|---|---|---|---|---|---|---|---|---|---|---|---|
| 28 | YH3–8 | 3773.7 | Shale | Longmaxi | LM4 | 32.0 | 4.0 | 6.0 | 51.0 | / | 7.0 | 5.0 | 2.49 |
| 29 | YH3–8 | 3775.8 | Shale | Longmaxi | LM4 | 37.0 | 4.0 | 9.0 | 46.0 | / | 10.0 | 5.0 | 2.11 |
| 30 | Z204 | 3406.8 | Shale | Longmaxi | LM2–3 | 46.0 | 2.0 | 11.0 | 35.0 | / | 11.0 | 3.0 | 1.73 |
| 31 | XD2 | 2065.4 | Shale | Longmaxi | LM2–3 | 41.0 | 4.0 | 4.0 | 27.0 | / | 8.0 | 4.0 | 2.43 |
| 32 | XD2 | 2067.8 | Shale | Longmaxi | LM2–3 | 29.0 | 3.0 | 28.0 | 39.0 | / | 5.0 | 3.0 | 1.70 |
| 33 | Y105 | 1685.0 | Shale | Longmaxi | LM2–3 | 36.0 | 7.0 | 10.0 | 32.0 | / | 8.0 | 3.0 | 1.32 |
| 34 | Y105 | 1687.0 | Shale | Longmaxi | LM2–3 | 47.0 | 6.0 | 8.0 | 36.0 | / | 9.0 | 3.0 | 2.09 |
| 35 | Y105 | 1688.2 | Shale | Longmaxi | LM2–3 | 35.0 | 4.0 | 6.0 | 27.0 | / | 10.0 | 4.0 | 1.06 |
| 36 | YH3–8 | 3779.1 | Shale | Longmaxi | LM2–3 | 37.0 | 6.0 | 9.0 | 41.0 | / | 12.0 | 3.0 | 2.05 |
| 37 | YH3–8 | 3781.0 | Shale | Longmaxi | LM2–3 | 43.0 | 3.0 | 7.0 | 33.0 | / | 7.0 | 4.0 | 2.69 |
| 38 | SH outcrop | / | Shale | Longmaxi | LM2–3 | 38.0 | 9.0 | 7.0 | 36.0 | / | 10.0 | 4.0 | 2.14 |
| 39 | SH outcrop | / | Shale | Longmaxi | LM2–3 | 47.0 | 7.0 | 7.0 | 32.0 | / | 9.0 | 3.0 | 2.10 |
| 40 | SH outcrop | / | Shale | Longmaxi | LM2–3 | 44.0 | 6.0 | 8.0 | 27.0 | / | 9.0 | 3.0 | 2.19 |
| 41 | SH outcrop | / | Shale | Longmaxi | LM2–3 | 48.0 | 5.0 | 4.0 | 30.0 | / | 10.0 | 3.0 | 2.22 |
| 42 | SH outcrop | / | Shale | Longmaxi | LM2–3 | 47.0 | 4.0 | 6.0 | 30.0 | / | 11.0 | 3.0 | 2.73 |
| 43 | XD2 | 2068.5 | Shale | Longmaxi | LM1 | 61.0 | 3.0 | 10.0 | 29.0 | / | 5.0 | 4.0 | 3.14 |
| 44 | XD2 | 2069.0 | Shale | Longmaxi | LM1 | 63.0 | 2.0 | 4.0 | 23.0 | / | 4.0 | 4.0 | 3.32 |
| 45 | XD2 | 2069.3 | Shale | Longmaxi | LM1 | 53.0 | 2.0 | 6.0 | 30.0 | / | 6.0 | 3.0 | 3.16 |
| 46 | Y105 | 1688.9 | Shale | Longmaxi | LM1 | 48.0 | 2.0 | 4.0 | 31.0 | / | 7.0 | 8.0 | 6.03 |
| 47 | Y105 | 1689.3 | Shale | Longmaxi | LM1 | 71.0 | 4.0 | 7.0 | 12.0 | / | 3.0 | 3.0 | 4.21 |
| 48 | Y105 | 1689.8 | Shale | Longmaxi | LM1 | 69.0 | 3.0 | 10.0 | 12.0 | / | 3.0 | 3.0 | 4.97 |
| 49 | YH3–8 | 3782.2 | Shale | Longmaxi | LM1 | 56.0 | 8.0 | 15.0 | 14.0 | / | 4.0 | 3.0 | 4.76 |
| 50 | SH outcrop | / | Shale | Longmaxi | LM1 | 65.0 | 7.0 | 13.0 | 9.0 | / | 3.0 | 3.0 | 5.21 |

**Table 1.** *Cont.*

| No. | Well | Depth | Lithology | Formation | Graptolite Zone | Quartz | Calcite | Dolomite | Clay Minerals | Potassium Feldspar | Plagioclase | Pyrite | TOC |
|-----|------|-------|-----------|-----------|-----------------|--------|---------|----------|---------------|--------------------|-------------|--------|-----|
| 51 | SH outcrop | / | Shale | Longmaxi | LM1 | 73.0 | 4.0 | 5.0 | 10.0 | / | 6.0 | 2.0 | 7.11 |
| 52 | XD2 | 2069.5 | Shale | Guanyingqiao | WF4 | 36.0 | 29.0 | 18.0 | 13.0 | / | 2.00 | 2.00 | 2.34 |
| 53 | XD2 | 2069.6 | Shale | Guanyingqiao | WF4 | 25.0 | 27.0 | 18.0 | 17.0 | 1.0 | 7.00 | 5.00 | 2.40 |
| 54 | XD2 | 2069.7 | Shale | Guanyingqiao | WF4 | 25.0 | 32.0 | 18.0 | 16.0 | 1.0 | 3.00 | 5.00 | 1.97 |
| 55 | XD2 | 2069.9 | Shale | Guanyingqiao | WF4 | 22.0 | 32.0 | 17.0 | 17.0 | 1.0 | 5.00 | 6.00 | 2.26 |
| 56 | Y105 | 1690.2 | Shale | Guanyingqiao | WF4 | 24.0 | 34.0 | 15.0 | 19.0 | 1.0 | 4.00 | 3.00 | 2.18 |
| 57 | SH outcrop | / | Shale | Guanyingqiao | WF4 | 23.0 | 32.0 | 15.0 | 22.0 | 1.0 | 5.00 | 2.00 | 5.90 |
| 58 | YH3–8 | 3783.0 | Shale | Guanyingqiao | WF4 | 23.0 | 35.0 | 16.0 | 17.0 | 1.0 | 5.00 | 3.00 | 3.70 |
| 59 | YH3–8 | 3783.7 | Shale | Guanyingqiao | WF4 | 23.0 | 33.0 | 16.0 | 21.0 | 1.0 | 4.00 | 2.00 | 4.30 |
| 60 | Z204 | 3409.6 | Shale | Wufeng | WF2–3 | 50.0 | 26.0 | 10.0 | 13.0 | / | 1.00 | / | 4.20 |
| 61 | XD2 | 2070.3 | Shale | Wufeng | WF2–3 | 48.0 | 26.0 | 11.0 | 14.0 | / | 1.00 | / | 4.40 |
| 62 | XD2 | 2072.9 | Shale | Wufeng | WF2–3 | 34.0 | 21.0 | 17.0 | 23.0 | 1.00 | 3.00 | 1.00 | 3.90 |
| 63 | XD2 | 2075.6 | Shale | Wufeng | WF2–3 | 60.0 | 16.0 | 9.0 | 12.0 | / | 1.00 | 2.00 | 3.90 |
| 64 | XD2 | 2078.8 | Shale | Wufeng | WF2–3 | 58.0 | 18.0 | 9.0 | 12.0 | / | 1.00 | 2.00 | 3.10 |
| 65 | XD2 | 2080.6 | Shale | Wufeng | WF2–3 | 55.0 | 17.0 | 15.0 | 13.0 | / | / | / | 3.20 |
| 66 | XD2 | 2081.8 | Shale | Wufeng | WF2–3 | 50.0 | 19.0 | 15.0 | 12.0 | 1.00 | 1.00 | 2.00 | 3.10 |
| 67 | Y105 | 1690.5 | Shale | Wufeng | WF2–3 | 47.0 | 22.0 | 18.0 | 12.0 | / | 1.00 | / | 3.00 |
| 68 | Y105 | 1691.2 | Shale | Wufeng | WF2–3 | 51.0 | 22.0 | 11.0 | 15.0 | / | 1.00 | / | 3.50 |
| 69 | Y105 | 1691.5 | Shale | Wufeng | WF2–3 | 58.0 | 19.0 | 9.0 | 13.0 | / | / | 1.00 | 2.60 |
| 70 | Y105 | 1691.9 | Shale | Wufeng | WF2–3 | 41.0 | 23.0 | 16.0 | 16.0 | 1.00 | 1.00 | 2.00 | 4.60 |
| 71 | YH3–8 | 3784.5 | Shale | Wufeng | WF2–3 | 56.0 | 14.0 | 13.0 | 14.0 | / | 1.00 | 2.00 | 3.20 |
| 72 | YH3–8 | 3788.2 | Shale | Wufeng | WF2–3 | 37.0 | 19.0 | 26.0 | 14.0 | 1.00 | 1.00 | 2.00 | 4.10 |
| 73 | YH3–8 | 3791.5 | Shale | Wufeng | WF2–3 | 42.0 | 23.0 | 20.0 | 12.0 | 1.00 | 1.00 | 1.00 | 3.10 |
| 74 | YH3–8 | 3792.6 | Shale | Wufeng | WF2–3 | 42.0 | 24.0 | 19.0 | 13.0 | / | 1.00 | 1.00 | 3.40 |
| 75 | SH outcrop | / | Shale | Wufeng | WF2–3 | 47.0 | 23.0 | 16.0 | 12.0 | / | 1.00 | 1.00 | 3.00 |

**Table 1.** *Cont.*

| No. | Well | Depth | Lithology | Formation | Graptolite Zone | Quartz | Calcite | Dolomite | Clay Minerals | Potassium Feldspar | Plagioclase | Pyrite | TOC |
|-----|------|-------|-----------|-----------|-----------------|--------|---------|----------|---------------|--------------------|-------------|--------|-----|
| 76 | SH outcrop | / | Shale | Wufeng | WF2–3 | 42.0 | 25.0 | 19.0 | 13.0 | / | / | 1.00 | 2.80 |
| 77 | SH outcrop | / | Shale | Wufeng | WF2–3 | 33.0 | 28.0 | 16.0 | 21.0 | / | 1.00 | 1.00 | 3.70 |
| 78 | SH outcrop | / | Shale | Wufeng | WF2–3 | 40.0 | 20.0 | 26.0 | 13.0 | / | / | 1.00 | 4.00 |
| 79 | SH outcrop | / | Shale | Wufeng | WF2–3 | 37.0 | 20.0 | 27.0 | 12.0 | 1.00 | 1.00 | 2.00 | 3.60 |
| 80 | SH outcrop | / | Shale | Wufeng | WF2–3 | 33.0 | 19.0 | 34.0 | 12.0 | / | 1.00 | 1.00 | 3.30 |
| 81 | SH outcrop | / | Shale | Wufeng | WF2–3 | 27.0 | 24.0 | 35.0 | 10.0 | / | 2.00 | 2.00 | 2.50 |
| 82 | SH outcrop | / | Shale | Wufeng | WF2–3 | 30.0 | 25.0 | 35.0 | 10.0 | / | / | / | 2.40 |
| 83 | SH outcrop | / | Shale | Wufeng | WF2–3 | 25.0 | 27.0 | 34.0 | 12.0 | / | / | 2.00 | 2.50 |
| | | Average | | | | 39.94 | 12.29 | 11.75 | 28.31 | 0.16 | 5.19 | 2.66 | 2.64 |

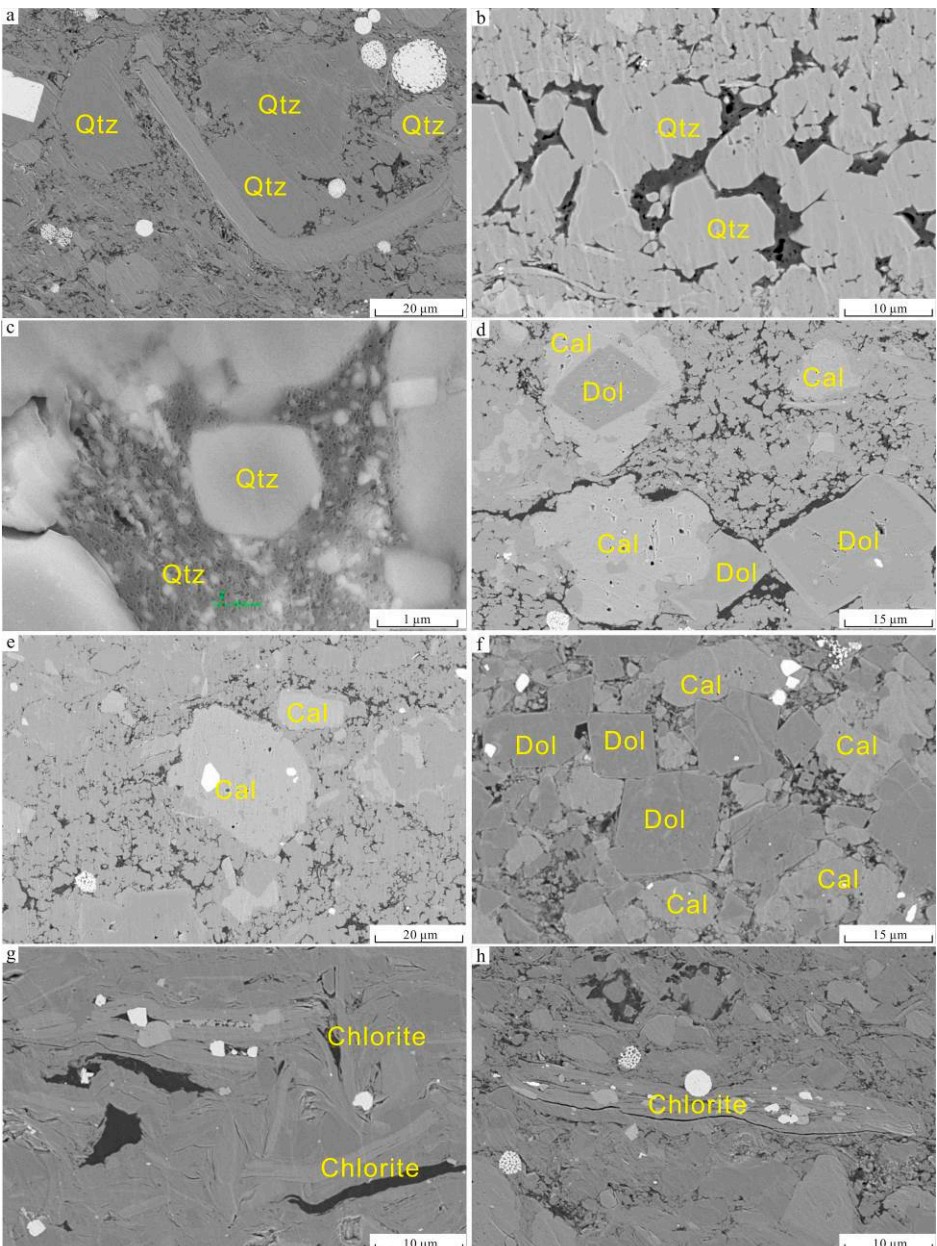

**Figure 4.** SEM photos showing the occurrence of typical minerals of the Wufeng–Longmaxi shale on the Yangtze Platform, South China. (**a**) Silt-sized quartz, well N211, 2334.17 m; (**b**) microcrystalline quartz, well YH3–8, 3783.4 m; (**c**) quartz associated with clay minerals; (**d**) calcite and dolomite, dissolution pores can be observed on the calcite surface, well YH3–8, 3784.49 m; (**e**) calcite, dissolution pores can be observed on the surface, well YH3–8, 3783.4 m; (**f**) calcite and dolomite, dissolution pores and crushing lines can be observed on the calcite surface, well YH3–8, 3783.4 m; (**g**) chlorite, well N211, 2321.05 m; (**h**) chlorite, well N211, 2334.17 m. Qtz: Quartz, Cal: Calcite, Dol: Dolomite.

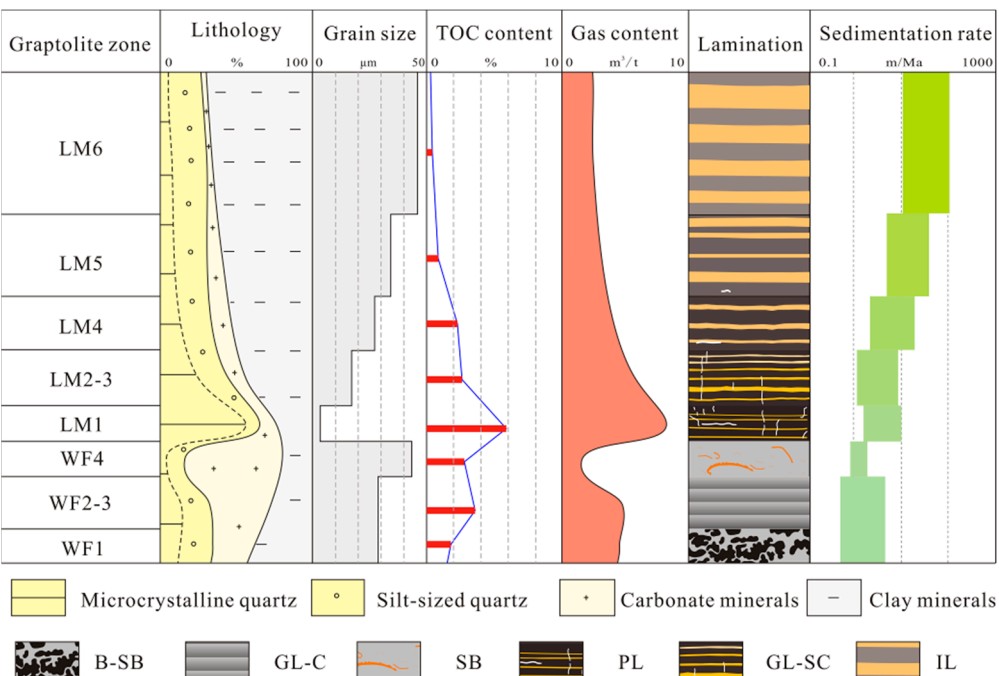

**Figure 5.** Characteristics of the sweet-spot interval of the Wufeng–Longmaxi shale on the Yangtze Platform, South China. This figure is modified from [29]. B–SB: Bioturbated-type massive bedding; GL–C: graded lamination composed of claystone; H–SB: heterogeneous-type massive bedding; PL: paper lamination; GL–SC: graded lamination composed of siltstone and claystone; IL: interlaminated lamination composed of siltstone and claystone.

The content of carbonate minerals ranges from 1% to 70.0% with an average of 24.04%. Carbonate minerals possibly originated from a terrigenous supply and/or primary chemical precipitation. The primary chemical precipitates of carbonate minerals mainly include calcite and dolomite. Both calcite and dolomite, in which a relatively large one has grain sizes reaching up to 20–40 μm, are dispersed among other minerals. The content of calcite ranges from 1% to 35.0%, with an average of 12.29%. Under SEM, calcite is relatively light in color and predominantly irregularly shaped (Figure 4d–f). Dissolution pores (Figure 4d,e), longitudinal stripes, and crushing lines (Figure 4f) can be observed on the surface. The content of dolomite ranges from 0 to 35.0%, with an average of 11.75%. Under SEM, dolomite is relatively dark and occurs mostly as regular euhedral crystals (Figure 4d,f). In addition, some semi-euhedral calcite crystals are surrounded by a lighter overgrowth edge (Figure 4e), which is the replacement of dolomite by ankerite, showing a distinct rhombic crystal shape and almost no dissolved pores. The contents of carbonate minerals are the highest in the WF4 interval and reach up to 31.8% on average, according to XRD analysis (Table 1; Figures 5 and 6).

The contents of clay minerals range from 9.0% to 54.0%, with an average of 28.31%. Clay minerals are dominated by illite, chlorite, and I/S, with minor amounts of kaolinite, according to SEM–EDX data. Illite is mostly in the form of flakes, needle clusters, or flocs and usually retains transportation traces on large-particle surfaces. Chlorite commonly has an elongated shape and loosened interlayers and is intercalated with euhedral pyrite particles. Chlorite is usually bent or deformed with the shape of the host rock and shows obvious plasticity (Figure 4g,h). I/S is mostly lamellar and has a honeycomb shape of aggregates. Quartz particles are usually embedded in or wrapped by I/S minerals (Figure 4c). For the Wufeng–Longmaxi shale, the contents of clay minerals increase from the bottom to the top and reach up to 49.9% in the LM6 interval (Figure 5).

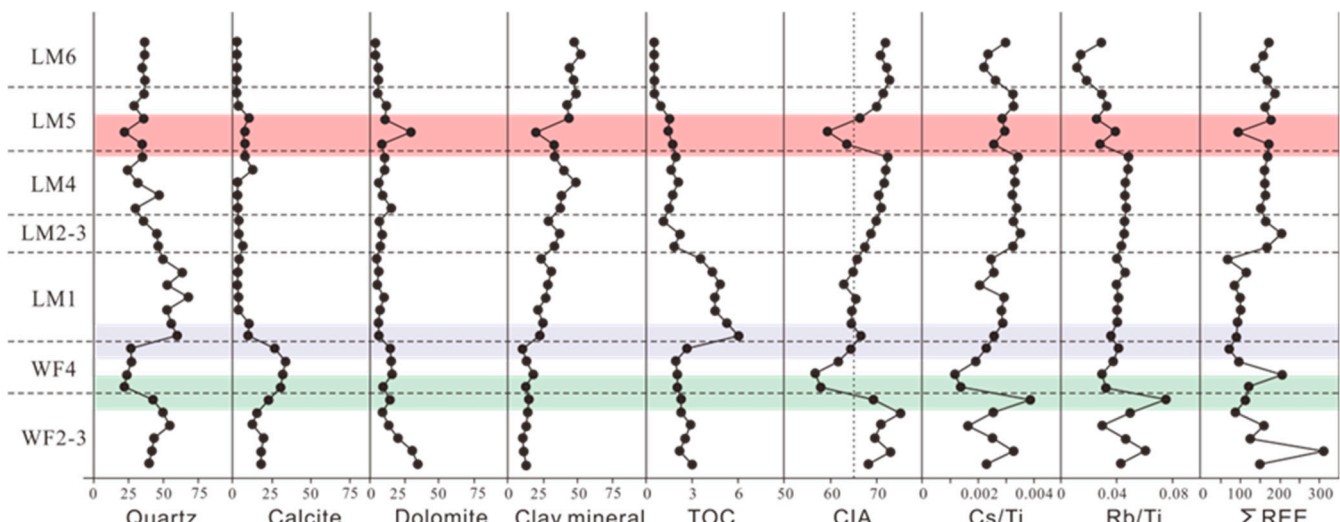

**Figure 6.** Variations in contents of some mineral compositions (%) [48], TOC contents (%), chemical index of alteration (CIA), Ti-normalized ratios, and total rare-earth elements (∑REE) showing the change in lithology and paleoclimate.

### 4.2. TOC Content and Dispersed Organic Matter

The content of TOC varies between 0.27% and 7.11%, with an average of 2.64%. The value is similar to that of the shales in southern Iran and the Arabian Plate, which have TOC contents ranging from 2% to 12% [9]. The dispersed organic matter is commonly pyrobitumen with minor amounts of vitrinite and fusinite. Pyrobitumen, which is black in reflected white light, commonly has void-filling and embayment textures (Figure 7a,b). Vitrinite, which is gray or pale gray in reflected white light, commonly has an elongated or irregular shape (Figure 7). Fusinite, which is white in reflected white light, commonly has an irregular or globular shape (Figure 7). The LM1 interval has the highest contents of TOC (avg. 4.6%) and sapropelic (reaching up to 100%). Moving upwards, the TOC content gradually decreases together with the diminishing content of sapropelic and increasing content of inertinite.

### 4.3. Lithology and Depositional Structures

The Wufeng Formation is mainly composed of graptolite-rich black shale with bioturbated-type structureless beds [29] (Figure 8a) or graded lamination composed of claystone (Figure 8b). In this interval, the grain sizes of calcite and dolomite vary between 22 and 45 μm (Figure 9a). The Guanyingqiao bed consists of bioclastic limestone or argillaceous limestone with massive-type structureless beds (Figure 8c,d). The mineral composition mainly consists of calcite and dolomite with small amounts of quartz (Figure 9b). The Longmaxi Formation consists of laterally extensive graptolite-rich, largely unbioturbated, gray to black shales, mudstone, siltstone, and sandwiched K–bentonite. From bottom to top, paper lamination (Figure 8e), graded lamination composed of claystone and siltstone (Figure 8f,g), and interlaminated lamination composed of siltstone and claystone occur in succession (Figure 8h). The LM1 interval predominantly consists of quartz with grain sizes less than 4 μm, and from LM1 to LM6, the carbonate mineral content and grain size gradually increase from the bottom to the top (Figure 9c–f).

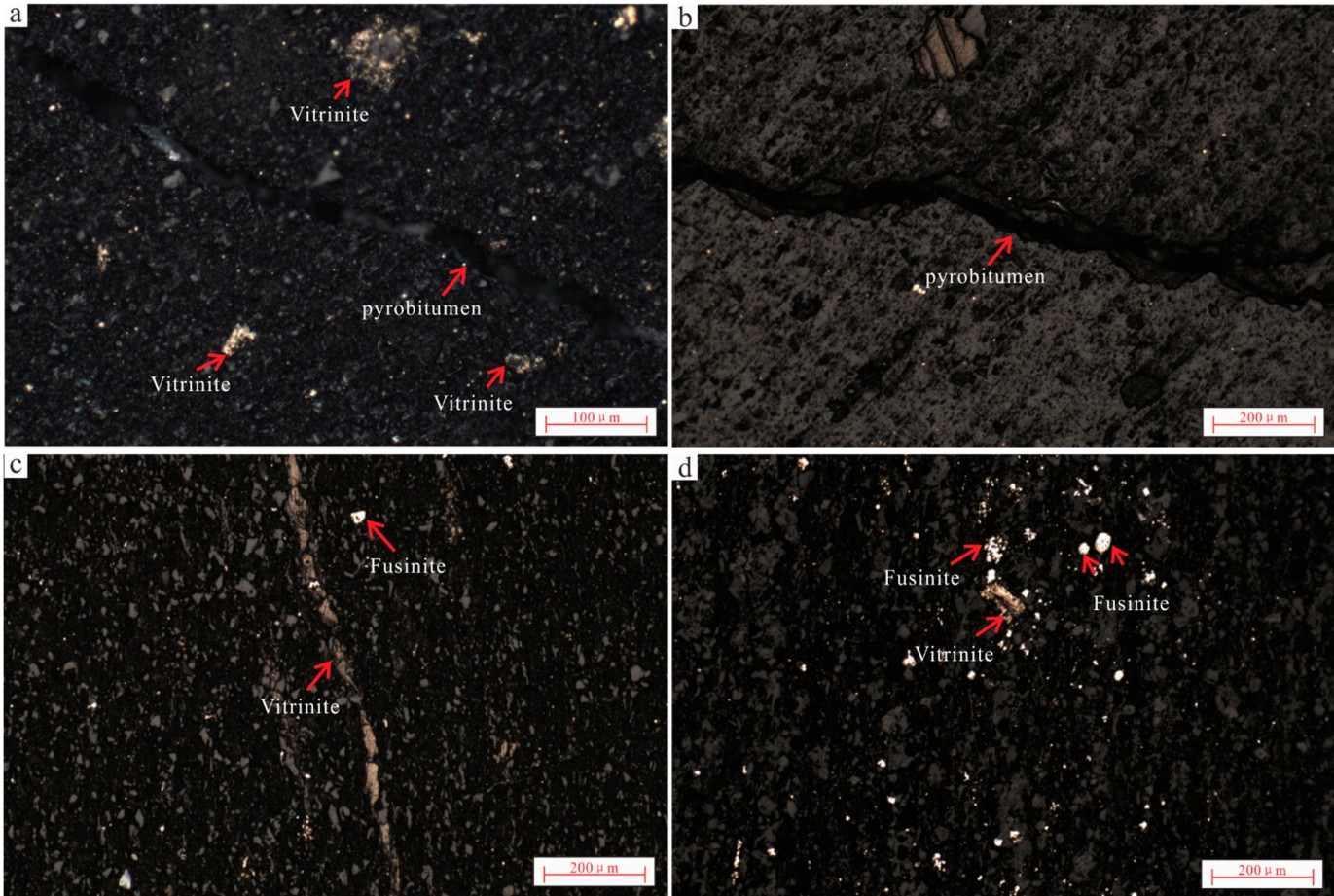

**Figure 7.** Photomicrographs of organic matter in reflected white light of the Wufeng–Longmaxi shale on the Yangtze Platform, South China. (**a**) Pyrobitumen in the image is black, and vitrinite is gray and irregularly shaped, sample number: YH-242; (**b**) pyrobitumen in the image is black, sample number: SH9-15-1; (**c**) vitrinite in the image is pale gray and has an elongated shaped, sample number: YH-242; (**d**) vitrinite in the image is pale gray and has an irregular or elongated shape, and fusinite is white and has an irregular or globular shape, sample number: SH9-15-1.

*4.4. Geochemistry*

Consistent with the mineral compositions, the XRF results show that $SiO_2$ and $Al_2O_3$ are the dominant major element oxides in the Wufeng–Longmaxi shale (Table 2). The $SiO_2$ content ranges from 20.56% to 78.69%, with an average of 56.11%. The contents of $Al_2O_3$, $Fe_2O_{3T}$ (referring to the total contents of $Fe_2O_3$ and $FeO$), and $K_2O$ vary between 1.95% and 20.55% (avg. 8.89%), 0.91% and 13.82% (avg. 3.46%), and 0.51% and 5.36% (avg. 2.31%), respectively. The $CaO$ content ranges from 0.83% to 33.17% (avg. 10.06%). Other major element oxides contents are relatively low. Generally, except for $MgO$, $MnO$, and $P_2O_5$, the average major element contents of the Wufeng–Longmaxi shale are dramatically different from PAAS and NAAS. In detail, the value of $SiO_2$ is significantly lower than that of PAAS and NAAS, those of $Al_2O_3$, $Fe_2O_{3T}$, $NaO$, and $TiO_2$ are less than the half values of PAAS and NAAS, and the value of $CaO$ is 3~8 times that of PAAS and NAAS.

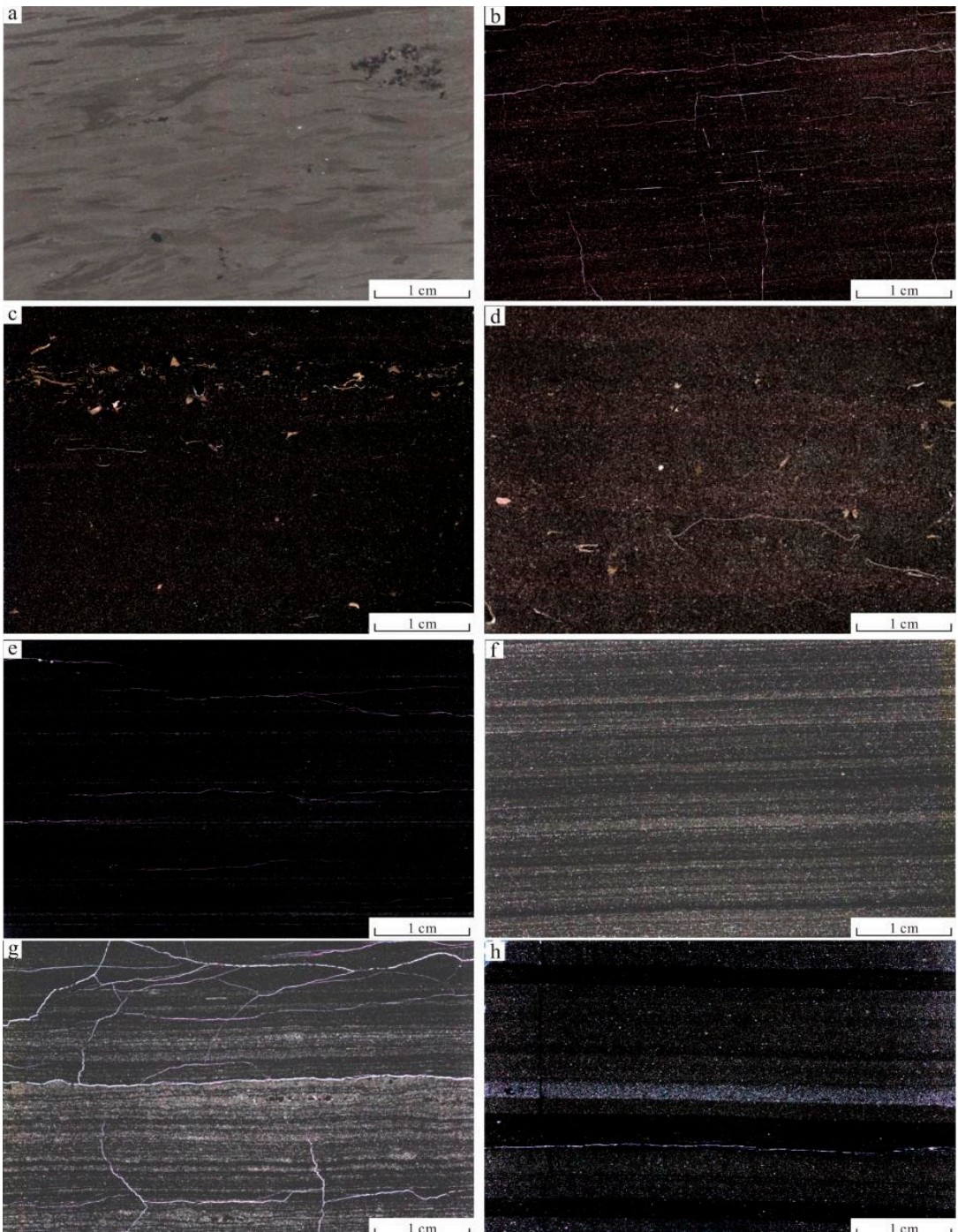

**Figure 8.** Typical sedimentologic characteristics of the Wufeng-Longmaxi shale on the Yangtze Platform, South China. (**a**) Bioturbated-type structureless beds, YH3–8, Wufeng Formation; (**b**) graded lamination composed of claystone, Chuanghe outcrop, Wufeng Formation; (**c**) massive-type structureless beds, Shuanghe outcrop, Guanyingqiao bed; (**d**) massive-type structureless beds, Shuanghe outcrop, Guanyingqiao bed; (**e**) paper lamination, Z204, Longmaxi Formation; (**f**,**g**) graded lamination composed of siltstone and claystone, N211, Longmaxi Formation; (**h**) interlaminated lamination composed of siltstone and claystone, Z204, Longmaxi Formation.

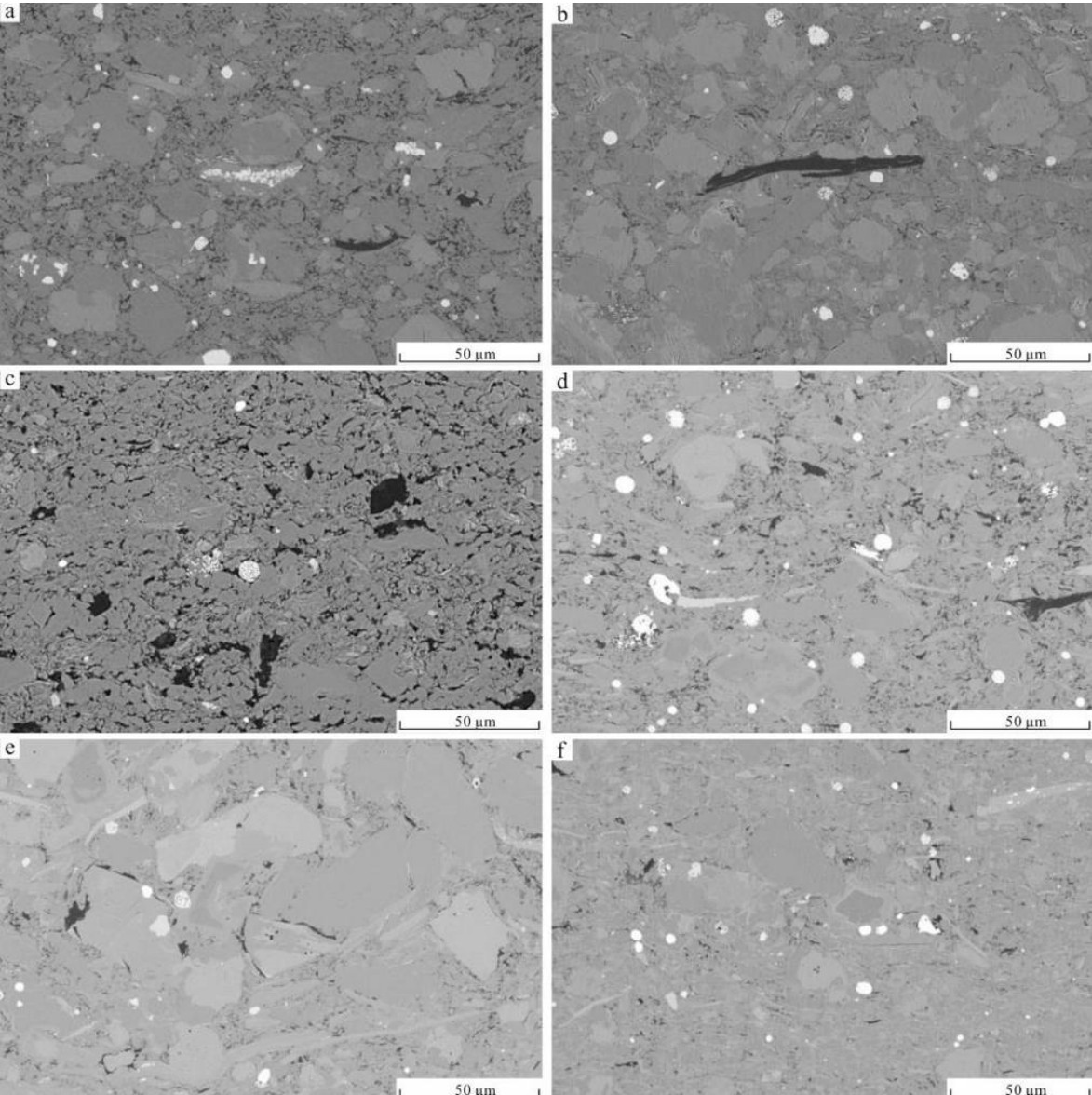

**Figure 9.** SEM photos showing typical minerals and grain sizes of various intervals of the Wufeng–Longmaxi shale on the Yangtze Platform, South China. (**a**) Sample from the Wufeng Formation of well N211, mainly composed of quartz, calcite, dolomite, and clay minerals with grain sizes ranging from 22 to 45 μm; (**b**) sample from the Guanyingqiao bed of well N211, mainly composed of calcite and dolomite with small amounts of quartz with grain sizes ranging from 20 to 40 μm; (**c**) sample from LM1 of well Z201, mainly composed of quartz with grain sizes less than 4 μm; (**d**) sample from LM2–3 of well YH3–8, mainly composed of quartz with small amounts of calcite and dolomite; (**e**) sample from LM4–5 of well YH3–8, mainly composed of quartz, calcite, and dolomite with small amounts of clay minerals; (**f**) sample from LM6 of well YH3–8, mainly composed of quartz, calcite, and dolomite with small amounts of clay minerals.

**Table 2.** Major element concentrations and organic carbon contents of the Wufeng–Longmaxi shale on the Yangtze Platform, South China.

| No. | Well | Depth | Lithology | Formation | Graptolite Zone | Sample No. | $SiO_2$ | $Al_2O_3$ | CaO | $Fe_2O_{3T}$ | $K_2O$ | MgO | MnO | $Na_2O$ | $P_2O_5$ | $TiO_2$ | LOI | TOTAL | $K_2O/Na_2O$ | $SiO_2/Al_2O_3$ | CIA | ICV |
|---|---|---|---|---|---|---|---|---|---|---|---|---|---|---|---|---|---|---|---|---|---|---|
| 1 | Y105 | 1650.5 | Shale | Longmaxi | LM6 | Y105–1 | 57.36 | 15.83 | 3.93 | 5.52 | 3.98 | 2.96 | 0.03 | 0.71 | 0.10 | 0.73 | 8.62 | 99.77 | 5.61 | 3.62 | 70.40 | 1.18 |
| 2 | Y105 | 1655.1 | Shale | Longmaxi | LM6 | Y105–11 | 61.76 | 12.14 | 5.08 | 4.01 | 3.29 | 2.42 | 0.03 | 0.56 | 0.11 | 0.59 | 9.71 | 99.70 | 5.88 | 5.09 | 69.16 | 1.23 |
| 3 | Y105 | 1658.4 | Shale | Longmaxi | LM6 | Y105–18 | 58.30 | 11.93 | 6.88 | 4.47 | 3.25 | 2.41 | 0.04 | 0.54 | 0.13 | 0.61 | 11.10 | 99.66 | 6.02 | 4.89 | 69.23 | 1.27 |
| 4 | YH3–8 | 3742.4 | Shale | Longmaxi | LM6 | YH–1 | 60.52 | 15.81 | 2.16 | 5.37 | 3.75 | 2.57 | 0.03 | 0.78 | 0.10 | 0.55 | 8.28 | 99.91 | 4.81 | 3.83 | 70.44 | 1.10 |
| 5 | YH3–8 | 3744.4 | Shale | Longmaxi | LM6 | YH–10 | 59.86 | 15.42 | 2.51 | 5.34 | 3.86 | 2.67 | 0.03 | 0.76 | 0.11 | 0.57 | 8.79 | 99.92 | 5.08 | 3.88 | 69.74 | 1.15 |
| 6 | YH3–8 | 3746.9 | Shale | Longmaxi | LM6 | YH–30 | 60.41 | 15.17 | 2.38 | 5.28 | 3.77 | 2.64 | 0.03 | 0.69 | 0.10 | 0.52 | 8.58 | 99.57 | 5.46 | 3.98 | 70.46 | 1.13 |
| 7 | YH3–8 | 3747.3 | Shale | Longmaxi | LM6 | YH–33 | 59.45 | 16.38 | 2.18 | 5.54 | 3.97 | 2.77 | 0.03 | 0.72 | 0.10 | 0.55 | 8.28 | 99.98 | 5.51 | 3.63 | 71.04 | 1.10 |
| 8 | XD2 | 2052.5 | Shale | Longmaxi | LM5 | XD–B54 | 53.24 | 13.05 | 5.73 | 4.25 | 3.59 | 3.00 | 0.03 | 0.64 | 0.08 | 0.53 | 13.31 | 97.45 | 5.61 | 4.08 | 68.50 | 1.31 |
| 9 | XD2 | 2055.3 | Shale | Longmaxi | LM5 | XD–B50 | 41.93 | 8.70 | 14.61 | 3.15 | 2.52 | 5.31 | 0.07 | 0.36 | 0.08 | 0.40 | 21.22 | 98.35 | 7.00 | 4.82 | 68.94 | 2.31 |
| 10 | Y105 | 1664.1 | Shale | Longmaxi | LM5 | Y105–31 | 60.23 | 11.60 | 6.81 | 3.61 | 3.26 | 2.27 | 0.03 | 0.48 | 0.10 | 0.50 | 11.10 | 99.99 | 6.79 | 5.19 | 69.39 | 1.20 |
| 11 | Y105 | 1668.1 | Shale | Longmaxi | LM5 | Y105–40 | 55.73 | 10.10 | 8.43 | 5.00 | 2.76 | 3.19 | 0.07 | 0.47 | 0.13 | 0.54 | 13.40 | 99.82 | 5.87 | 5.52 | 68.98 | 1.65 |
| 12 | Y105 | 1673.1 | Shale | Longmaxi | LM5 | Y105–52 | 63.46 | 9.58 | 6.78 | 3.23 | 2.33 | 2.05 | 0.03 | 0.45 | 0.12 | 0.47 | 11.22 | 99.72 | 5.18 | 6.62 | 70.50 | 1.25 |
| 13 | YH3–8 | 3749.4 | Shale | Longmaxi | LM5 | YH–44 | 58.84 | 15.75 | 2.92 | 5.46 | 3.89 | 2.82 | 0.03 | 0.78 | 0.09 | 0.55 | 8.62 | 99.75 | 4.99 | 3.74 | 69.88 | 1.16 |
| 14 | YH3–8 | 3753.4 | Shale | Longmaxi | LM5 | YH–67 | 61.23 | 14.31 | 3.01 | 5.02 | 3.51 | 2.77 | 0.04 | 0.81 | 0.09 | 0.53 | 8.62 | 99.94 | 4.33 | 4.28 | 68.85 | 1.22 |
| 15 | YH3–8 | 3757.1 | Shale | Longmaxi | LM5 | YH–85 | 56.42 | 14.75 | 4.66 | 5.11 | 3.48 | 3.27 | 0.05 | 1.16 | 0.11 | 0.59 | 10.26 | 99.85 | 3.00 | 3.83 | 66.02 | 1.36 |
| 16 | YH3–8 | 3761.7 | Shale | Longmaxi | LM5 | YH–117 | 25.33 | 3.84 | 33.17 | 3.03 | 0.76 | 3.08 | 0.39 | 0.51 | 0.07 | 0.15 | 29.42 | 99.75 | 1.49 | 6.60 | 60.54 | 3.40 |
| 17 | YH3–8 | 3766.0 | Shale | Longmaxi | LM5 | YH–142 | 59.51 | 13.06 | 5.03 | 4.93 | 3.02 | 2.54 | 0.04 | 1.26 | 0.11 | 0.59 | 9.88 | 99.98 | 2.40 | 4.56 | 63.76 | 1.37 |
| 18 | Z204 | 3398.6 | Shale | Longmaxi | LM4 | Z4–23 | 62.41 | 18.43 | 0.83 | 6.62 | 4.98 | 2.93 | 0.06 | 0.31 | 0.10 | 0.64 | 1.28 | 98.60 | 15.83 | 3.39 | 74.09 | 1.03 |
| 19 | XD2 | 2058.6 | Shale | Longmaxi | LM4 | XD–B44 | 43.03 | 20.55 | 4.67 | 8.95 | 5.36 | 2.91 | 0.08 | 0.56 | 0.03 | 0.34 | 11.69 | 98.17 | 9.57 | 2.09 | 72.85 | 1.04 |
| 20 | XD2 | 2061.1 | Shale | Longmaxi | LM4 | XD–B41 | 41.49 | 5.44 | 18.85 | 1.97 | 1.58 | 5.19 | 0.05 | 0.26 | 0.05 | 0.26 | 23.36 | 98.50 | 6.08 | 7.63 | 67.92 | 3.21 |
| 21 | XD2 | 2062.6 | Shale | Longmaxi | LM4 | XD–B38 | 42.42 | 6.24 | 17.27 | 2.03 | 1.82 | 5.57 | 0.05 | 0.29 | 0.05 | 0.30 | 22.98 | 99.02 | 6.28 | 6.80 | 68.05 | 3.03 |
| 22 | Y105 | 1674.6 | Shale | Longmaxi | LM4 | Y105–58 | 60.26 | 12.14 | 6.02 | 3.75 | 3.03 | 2.41 | 0.03 | 0.52 | 0.11 | 0.54 | 11.14 | 99.95 | 5.83 | 4.96 | 70.83 | 1.18 |
| 23 | Y105 | 1676.3 | Shale | Longmaxi | LM4 | Y105–64 | 60.23 | 10.87 | 6.84 | 4.08 | 2.74 | 2.04 | 0.03 | 0.49 | 0.13 | 0.52 | 11.49 | 99.46 | 5.59 | 5.54 | 70.33 | 1.20 |
| 24 | Y105 | 1677.7 | Shale | Longmaxi | LM4 | Y105–69 | 60.54 | 10.28 | 7.15 | 4.22 | 2.56 | 2.03 | 0.03 | 0.48 | 0.13 | 0.51 | 11.67 | 99.60 | 5.33 | 5.89 | 70.23 | 1.26 |
| 25 | Y105 | 1679.5 | Shale | Longmaxi | LM4 | Y105–75 | 59.89 | 9.89 | 7.55 | 3.62 | 2.53 | 2.33 | 0.04 | 0.50 | 0.13 | 0.50 | 12.77 | 99.75 | 5.06 | 6.06 | 69.26 | 1.35 |
| 26 | Y105 | 1681.8 | Shale | Longmaxi | LM4 | Y105–82 | 61.83 | 10.69 | 6.10 | 3.68 | 2.76 | 2.07 | 0.03 | 0.49 | 0.12 | 0.51 | 11.99 | 100.27 | 5.63 | 5.78 | 69.88 | 1.21 |
| 27 | YH3–8 | 3769.8 | Shale | Longmaxi | LM4 | YH–169 | 67.83 | 9.98 | 2.31 | 4.83 | 2.38 | 1.74 | 0.03 | 0.59 | 0.10 | 0.36 | 9.81 | 99.95 | 4.03 | 6.80 | 68.81 | 1.26 |

**Table 2.** *Cont.*

| No. | Well | Depth | Lithology | Formation | Graptolite Zone | Sample No. | $SiO_2$ | $Al_2O_3$ | CaO | $Fe_2O_{3T}$ | $K_2O$ | MgO | MnO | $Na_2O$ | $P_2O_5$ | $TiO_2$ | LOI | TOTAL | $K_2O/Na_2O$ | $SiO_2/Al_2O_3$ | CIA | ICV |
|---|---|---|---|---|---|---|---|---|---|---|---|---|---|---|---|---|---|---|---|---|---|---|
| 28 | YH3–8 | 3773.7 | Shale | Longmaxi | LM4 | YH–190 | 66.41 | 9.17 | 3.60 | 3.97 | 2.15 | 1.81 | 0.03 | 0.59 | 0.15 | 0.35 | 11.65 | 99.88 | 3.64 | 7.24 | 68.21 | 1.30 |
| 29 | YH3–8 | 3775.8 | Shale | Longmaxi | LM4 | YH–202 | 72.10 | 7.47 | 3.05 | 2.74 | 1.70 | 1.37 | 0.03 | 0.50 | 0.06 | 0.26 | 10.59 | 99.87 | 3.40 | 9.65 | 68.16 | 1.22 |
| 30 | Z204 | 3406.8 | Shale | Longmaxi | LM2–3 | Z4–8 | 48.04 | 12.24 | 13.50 | 3.75 | 2.80 | 3.73 | 0.31 | 0.35 | 0.09 | 0.42 | 2.97 | 88.20 | 7.93 | 3.93 | 74.48 | 1.40 |
| 31 | XD2 | 2065.4 | Shale | Longmaxi | LM2–3 | XD–B35 | 39.90 | 5.43 | 21.80 | 1.81 | 1.62 | 3.73 | 0.06 | 0.24 | 0.05 | 0.25 | 23.84 | 98.73 | 6.75 | 7.35 | 68.07 | 2.51 |
| 32 | XD2 | 2067.8 | Shale | Longmaxi | LM2–3 | XD–B32 | 69.10 | 3.47 | 9.87 | 0.91 | 1.00 | 1.13 | 0.03 | 0.19 | 0.05 | 0.16 | 13.28 | 99.19 | 5.26 | 19.91 | 66.98 | 1.56 |
| 33 | Y105 | 1685.0 | Shale | Longmaxi | LM2–3 | Y105–92 | 57.51 | 9.64 | 7.43 | 3.69 | 2.46 | 2.51 | 0.04 | 0.51 | 0.13 | 0.49 | 13.86 | 98.27 | 4.82 | 5.97 | 68.92 | 1.43 |
| 34 | Y105 | 1687.0 | Shale | Longmaxi | LM2–3 | Y105–98 | 58.72 | 10.55 | 6.71 | 3.21 | 2.70 | 2.70 | 0.03 | 0.65 | 0.10 | 0.52 | 14.09 | 99.98 | 4.15 | 5.57 | 67.55 | 1.39 |
| 35 | Y105 | 1688.2 | Shale | Longmaxi | LM2–3 | Y105–102 | 50.30 | 8.68 | 11.00 | 3.27 | 2.17 | 3.97 | 0.06 | 0.60 | 0.11 | 0.46 | 19.05 | 99.67 | 3.62 | 5.79 | 66.72 | 1.98 |
| 36 | YH3–8 | 3779.1 | Shale | Longmaxi | LM2–3 | YH–226 | 65.69 | 8.47 | 4.71 | 3.96 | 2.00 | 1.81 | 0.03 | 0.56 | 0.09 | 0.33 | 11.86 | 99.51 | 3.57 | 7.76 | 67.85 | 1.37 |
| 37 | YH3–8 | 3781.0 | Shale | Longmaxi | LM2–3 | YH–236 | 66.22 | 5.02 | 9.07 | 2.34 | 1.06 | 2.63 | 0.07 | 0.37 | 0.10 | 0.18 | 12.57 | 99.63 | 2.86 | 13.19 | 67.95 | 2.17 |
| 38 | SH outcrop | / | Shale | Longmaxi | LM2–3 | SH11–9–1 | 60.24 | 5.56 | 10.43 | 1.80 | 1.49 | 3.23 | 0.03 | 0.36 | 0.08 | 0.28 | 16.24 | 99.74 | 4.14 | 10.83 | 66.50 | 2.26 |
| 39 | SH outcrop | / | Shale | Longmaxi | LM2–3 | SH11–1–2 | 57.50 | 5.45 | 11.72 | 1.74 | 1.39 | 3.80 | 0.03 | 0.27 | 0.07 | 0.26 | 17.64 | 42.37 | 5.15 | 10.55 | 69.46 | 2.49 |
| 40 | SH outcrop | / | Shale | Longmaxi | LM2–3 | SH10–8–1 | 57.38 | 5.09 | 14.28 | 2.00 | 1.29 | 1.98 | 0.03 | 0.37 | 0.11 | 0.24 | 17.14 | 42.53 | 3.49 | 11.27 | 66.04 | 1.83 |
| 41 | SH outcrop | / | Shale | Longmaxi | LM2–3 | SH9–23–2 | 57.15 | 4.80 | 14.76 | 1.58 | 1.25 | 2.04 | 0.03 | 0.40 | 0.08 | 0.23 | 17.51 | 42.68 | 3.13 | 11.91 | 64.24 | 1.92 |
| 42 | SH outcrop | / | Shale | Longmaxi | LM2–3 | SH9–15–1 | 69.29 | 4.30 | 9.09 | 1.28 | 1.11 | 1.68 | 0.02 | 0.22 | 0.07 | 0.20 | 13.19 | 31.16 | 5.05 | 16.11 | 69.04 | 1.70 |
| 43 | XD2 | 2068.5 | Shale | Longmaxi | LM1 | XD–B31 | 75.11 | 4.57 | 4.48 | 1.21 | 1.31 | 1.22 | 0.02 | 0.30 | 0.05 | 0.20 | 10.84 | 99.31 | 4.37 | 16.44 | 65.49 | 1.44 |
| 44 | XD2 | 2069.0 | Shale | Longmaxi | LM1 | XD–B30 | 68.41 | 5.98 | 5.53 | 1.91 | 1.74 | 1.68 | 0.02 | 0.41 | 0.10 | 0.27 | 13.18 | 99.23 | 4.24 | 11.44 | 64.88 | 1.52 |
| 45 | XD2 | 2069.3 | Shale | Longmaxi | LM1 | XD–B29 | 44.40 | 3.39 | 20.11 | 1.47 | 1.04 | 3.06 | 0.06 | 0.25 | 0.10 | 0.16 | 25.33 | 99.37 | 4.16 | 13.10 | 63.47 | 3.24 |
| 46 | Y105 | 1688.9 | Shale | Longmaxi | LM1 | Y105–104 | 56.62 | 8.04 | 9.71 | 3.57 | 2.06 | 2.47 | 0.04 | 0.62 | 0.10 | 0.44 | 15.92 | 99.59 | 3.32 | 7.04 | 65.28 | 1.68 |
| 47 | Y105 | 1689.3 | Shale | Longmaxi | LM1 | Y105–106 | 60.75 | 7.20 | 8.04 | 4.02 | 1.82 | 2.28 | 0.04 | 0.59 | 0.11 | 0.40 | 14.40 | 99.65 | 3.08 | 8.44 | 64.77 | 1.79 |
| 48 | Y105 | 1689.8 | Shale | Longmaxi | LM1 | Y105–108 | 62.95 | 7.66 | 5.08 | 2.84 | 1.98 | 2.48 | 0.04 | 0.62 | 0.09 | 0.45 | 15.46 | 99.65 | 3.19 | 8.22 | 64.65 | 1.69 |
| 49 | YH3–8 | 3782.2 | Shale | Longmaxi | LM1 | YH–242 | 66.53 | 5.93 | 7.06 | 3.42 | 1.42 | 2.27 | 0.04 | 0.46 | 0.07 | 0.25 | 12.13 | 99.58 | 3.09 | 11.22 | 66.00 | 1.92 |
| 50 | SH outcrop | / | Shale | Longmaxi | LM1 | SH8–12–1 | 76.22 | 4.35 | 2.42 | 1.39 | 1.13 | 1.19 | 0.01 | 0.10 | 0.12 | 0.21 | 12.74 | 99.88 | 11.30 | 17.52 | 73.66 | 1.32 |
| 51 | SH outcrop | / | Shale | Longmaxi | LM1 | SH8–10–1 | 75.89 | 4.67 | 2.15 | 1.45 | 1.22 | 1.26 | 0.01 | 0.21 | 0.11 | 0.23 | 12.59 | 99.79 | 5.81 | 16.25 | 69.86 | 1.38 |
| 52 | XD2 | 2069.5 | Shale | Guanyingqiao | WF4 | XD–B28 | 50.43 | 6.88 | 16.74 | 1.72 | 2.01 | 2.31 | 0.06 | 0.48 | 0.10 | 0.35 | 17.69 | 98.77 | 4.19 | 7.33 | 64.66 | 1.64 |

**Table 2.** *Cont.*

| No. | Well | Depth | Lithology | Formation | Graptolite Zone | Sample No. | SiO$_2$ | Al$_2$O$_3$ | CaO | Fe$_2$O$_{3T}$ | K$_2$O | MgO | MnO | Na$_2$O | P$_2$O$_5$ | TiO$_2$ | LOI | TOTAL | K$_2$O/Na$_2$O | SiO$_2$/Al$_2$O$_3$ | CIA | ICV |
|---|---|---|---|---|---|---|---|---|---|---|---|---|---|---|---|---|---|---|---|---|---|---|
| 53 | XD2 | 2069.6 | Shale | Guanyingqiao | WF4 | XD–B27 | 44.73 | 6.15 | 18.26 | 3.99 | 1.81 | 2.61 | 0.07 | 0.52 | 0.14 | 0.32 | 18.70 | 97.30 | 3.48 | 7.27 | 62.60 | 2.18 |
| 54 | XD2 | 2069.7 | Shale | Guanyingqiao | WF4 | XD–B26 | 57.88 | 8.88 | 2.28 | 13.82 | 2.61 | 0.91 | 0.06 | 1.07 | 0.94 | 0.49 | 9.53 | 98.47 | 2.44 | 6.52 | 58.30 | 2.05 |
| 55 | XD2 | 2069.9 | Shale | Guanyingqiao | WF4 | XD–B24 | 58.11 | 9.36 | 4.94 | 9.21 | 2.74 | 1.33 | 0.06 | 1.07 | 0.30 | 0.50 | 10.32 | 97.94 | 2.56 | 6.21 | 59.04 | 1.76 |
| 56 | Y105 | 1690.2 | Shale | Guanyingqiao | WF4 | Y105–109 | 53.82 | 6.68 | 14.72 | 3.26 | 1.70 | 1.61 | 0.07 | 0.50 | 0.15 | 0.38 | 16.82 | 99.71 | 3.40 | 8.06 | 65.68 | 1.54 |
| 57 | SH outcrop | / | Shale | Guanyingqiao | WF4 | SH7–4–1 | 42.89 | 7.26 | 20.46 | 3.75 | 1.91 | 3.00 | 0.09 | 0.55 | 0.12 | 0.36 | 19.52 | 99.91 | 3.47 | 5.91 | 65.16 | 2.00 |
| 58 | YH3–8 | 3783.0 | Shale | Guanyingqiao | WF4 | YH–247 | 42.59 | 9.83 | 12.54 | 4.61 | 2.46 | 6.02 | 0.10 | 0.67 | 0.10 | 0.42 | 20.16 | 99.50 | 3.67 | 4.33 | 66.85 | 2.43 |
| 59 | YH3–8 | 3783.7 | Shale | Guanyingqiao | WF4 | YH–250 | 61.80 | 9.29 | 7.44 | 3.13 | 2.29 | 2.24 | 0.04 | 0.49 | 0.12 | 0.34 | 12.76 | 99.94 | 4.67 | 6.65 | 69.39 | 1.32 |
| 60 | Z204 | 3409.6 | Shale | Wufeng | WF2–3 | Z4–1 | 35.25 | 12.22 | 20.41 | 3.53 | 3.05 | 2.64 | 0.48 | 0.33 | 0.82 | 0.37 | 5.57 | 84.67 | 9.17 | 2.89 | 73.47 | 1.19 |
| 61 | XD2 | 2070.3 | Shale | Wufeng | WF2–3 | XD–B22 | 44.14 | 14.39 | 12.25 | 1.96 | 4.12 | 3.70 | 0.03 | 0.63 | 0.09 | 0.35 | 17.40 | 99.06 | 6.54 | 3.07 | 68.74 | 1.23 |
| 62 | XD2 | 2072.9 | Shale | Wufeng | WF2–3 | XD–B19 | 51.57 | 6.05 | 17.55 | 1.41 | 1.46 | 2.15 | 0.07 | 0.21 | 0.05 | 0.22 | 18.66 | 99.40 | 6.95 | 8.52 | 72.67 | 1.49 |
| 63 | XD2 | 2075.6 | Shale | Wufeng | WF2–3 | XD–B15 | 36.01 | 7.10 | 21.78 | 5.37 | 2.11 | 3.12 | 0.13 | 0.24 | 0.14 | 0.52 | 21.41 | 97.93 | 8.79 | 5.07 | 69.75 | 2.16 |
| 64 | XD2 | 2078.8 | Shale | Wufeng | WF2–3 | XD–B10 | 57.67 | 8.97 | 9.42 | 2.13 | 2.73 | 3.08 | 0.07 | 0.34 | 0.07 | 0.43 | 14.05 | 98.96 | 8.03 | 6.43 | 68.73 | 1.55 |
| 65 | XD2 | 2080.6 | Shale | Wufeng | WF2–3 | XD–B6 | 42.59 | 12.86 | 14.87 | 2.88 | 3.58 | 3.53 | 0.11 | 0.41 | 0.65 | 0.43 | 16.27 | 98.18 | 8.73 | 3.31 | 71.07 | 1.30 |
| 66 | XD2 | 2081.8 | Shale | Wufeng | WF2–3 | XD–B2 | 43.63 | 13.55 | 13.84 | 2.91 | 3.98 | 3.89 | 0.09 | 0.67 | 0.05 | 0.65 | 15.31 | 98.57 | 5.94 | 3.22 | 67.50 | 1.42 |
| 67 | Y105 | 1690.5 | Shale | Wufeng | WF2–3 | Y105–110 | 31.02 | 1.95 | 31.19 | 1.65 | 0.51 | 1.68 | 0.11 | 0.41 | 0.03 | 0.11 | 30.92 | 99.58 | 1.24 | 15.91 | 50.62 | 3.86 |
| 68 | Y105 | 1691.2 | Shale | Wufeng | WF2–3 | Y105–112 | 54.67 | 6.77 | 11.63 | 2.88 | 1.78 | 2.29 | 0.04 | 0.41 | 0.11 | 0.37 | 19.14 | 100.09 | 4.34 | 8.08 | 67.36 | 1.70 |
| 69 | Y105 | 1691.5 | Shale | Wufeng | WF2–3 | Y105–113 | 52.49 | 7.60 | 9.86 | 2.91 | 2.06 | 3.71 | 0.07 | 0.35 | 0.09 | 0.42 | 20.08 | 99.64 | 5.89 | 6.91 | 69.17 | 2.02 |
| 70 | Y105 | 1691.9 | Shale | Wufeng | WF2–3 | Y105–G1 | 60.54 | 11.88 | 2.29 | 3.66 | 3.30 | 3.00 | 0.03 | 0.46 | 0.14 | 0.64 | 13.96 | 99.90 | 7.17 | 5.10 | 69.99 | 1.34 |
| 71 | YH3–8 | 3784.5 | Shale | Wufeng | WF2–3 | YH–255 | 71.70 | 8.22 | 2.36 | 2.91 | 1.89 | 1.75 | 0.03 | 0.48 | 0.10 | 0.31 | 10.05 | 99.80 | 3.94 | 8.72 | 69.37 | 1.26 |
| 72 | YH3–8 | 3788.2 | Shale | Wufeng | WF2–3 | YH–270 | 50.25 | 10.66 | 10.91 | 4.69 | 2.83 | 4.27 | 0.22 | 0.59 | 0.29 | 0.49 | 14.70 | 99.90 | 4.80 | 4.71 | 68.02 | 1.86 |
| 73 | YH3–8 | 3791.5 | Shale | Wufeng | WF2–3 | YH–288 | 63.89 | 12.03 | 5.06 | 4.44 | 2.97 | 2.36 | 0.09 | 0.69 | 0.06 | 0.53 | 7.76 | 99.87 | 4.30 | 5.31 | 68.65 | 1.26 |
| 74 | YH3–8 | 3792.6 | Shale | Wufeng | WF2–3 | YH–295 | 20.56 | 4.85 | 30.95 | 2.05 | 1.02 | 1.18 | 0.18 | 0.39 | 0.05 | 0.22 | 38.55 | 99.98 | 2.62 | 4.24 | 66.99 | 1.49 |
| 75 | SH outcrop | / | Shale | Wufeng | WF2–3 | SH6–7–2 | 54.25 | 8.44 | 12.35 | 2.99 | 2.14 | 2.31 | 0.04 | 0.75 | 0.18 | 0.46 | 16.11 | 100.02 | 2.85 | 6.43 | 63.80 | 1.57 |
| 76 | SH outcrop | / | Shale | Wufeng | WF2–3 | SH6–3–1 | 59.61 | 9.25 | 8.47 | 2.27 | 2.33 | 2.07 | 0.03 | 0.52 | 0.16 | 0.46 | 14.70 | 99.87 | 4.48 | 6.44 | 68.57 | 1.25 |

**Table 2.** *Cont.*

| No. | Well | Depth | Lithology | Formation | Graptolite Zone | Sample No. | SiO$_2$ | Al$_2$O$_3$ | CaO | Fe$_2$O$_{3T}$ | K$_2$O | MgO | MnO | Na$_2$O | P$_2$O$_5$ | TiO$_2$ | LOI | TOTAL | K$_2$O/Na$_2$O | SiO$_2$/Al$_2$O$_3$ | CIA | ICV |
|---|---|---|---|---|---|---|---|---|---|---|---|---|---|---|---|---|---|---|---|---|---|---|
| 77 | SH outcrop | / | Shale | Wufeng | WF2–3 | SH5–33–2 | 39.58 | 4.57 | 20.60 | 2.78 | 1.21 | 3.91 | 0.06 | 0.34 | 0.13 | 0.23 | 26.66 | 100.07 | 3.56 | 8.66 | 65.27 | 3.18 |
| 78 | SH outcrop | / | Shale | Wufeng | WF2–3 | SH5–30–2 | 52.08 | 3.89 | 17.48 | 1.51 | 1.03 | 2.16 | 0.06 | 0.19 | 0.10 | 0.18 | 21.41 | 100.09 | 5.42 | 13.39 | 69.06 | 2.19 |
| 79 | SH outcrop | / | Shale | Wufeng | WF2–3 | SH5–26–2 | 57.34 | 3.82 | 15.63 | 1.37 | 1.06 | 1.70 | 0.05 | 0.18 | 0.08 | 0.18 | 18.45 | 99.86 | 5.89 | 15.01 | 68.67 | 1.90 |
| 80 | SH outcrop | / | Shale | Wufeng | WF2–3 | SH5–20–1 | 60.96 | 3.82 | 14.01 | 1.28 | 1.05 | 1.45 | 0.04 | 0.23 | 0.08 | 0.18 | 16.74 | 99.84 | 4.57 | 15.96 | 66.83 | 1.75 |
| 81 | SH outcrop | / | Shale | Wufeng | WF2–3 | SH4–32–1 | 78.69 | 4.44 | 3.61 | 1.42 | 1.24 | 1.16 | 0.02 | 0.11 | 0.07 | 0.22 | 9.01 | 99.99 | 11.27 | 17.72 | 72.22 | 1.32 |
| 82 | SH outcrop | / | Shale | Wufeng | WF2–3 | SH4–24–1 | 67.52 | 3.23 | 11.11 | 1.41 | 0.88 | 1.97 | 0.05 | 0.17 | 0.05 | 0.15 | 13.62 | 100.16 | 5.18 | 20.90 | 68.08 | 2.38 |
| 83 | SH outcrop | / | Shale | Wufeng | WF2–3 | SH4–16–1 | 63.13 | 4.02 | 11.31 | 1.55 | 1.15 | 2.71 | 0.05 | 0.12 | 0.06 | 0.19 | 15.45 | 99.74 | 9.58 | 15.70 | 70.99 | 2.45 |
| | | | | Average | | | 56.11 | 8.89 | 10.07 | 3.46 | 2.31 | 2.60 | 0.06 | 0.49 | 0.13 | 0.39 | 14.68 | 96.27 | 5.22 | 7.88 | 67.87 | 1.70 |
| | | | | PAAS | | | 62.8 | 18.80 | 1.29 | 6.50 | 3.68 | 2.19 | 0.11 | 1.19 | 0.16 | 0.99 | / | / | / | / | / | / |
| | | | | NASC | | | 64.8 | 16.90 | 3.63 | / | 3.97 | 2.86 | 0.06 | 1.14 | / | 0.70 | / | / | / | / | / | / |

Compared to the trace element contents of the upper continental crust (UCC) [49], several elements are remarkably enriched (Table S1; Figure 10), and they can be represented by the concentration coefficient (CC = the ratio of the element concentration in the Wufeng–Longmaxi shale vs. UCC) [50]. The trace elements Ba, Zn, and U have CCs ranging from 2 to 5, among which the trace element U has a CC very close to 5. Other elements, including Sr, Sc, Cr, Co, Ga, Ta, Nb, Rb, Th, Y, Tb, Gd, Eu, Sm, Ho, Er, Tm, Pr, Ce, La, Yb, Nd, Pb, Dy, Lu, Cs, V, Ni, Cu, and REEs, have CC values similar to the average of UCC ($0.5 < CC < 2$). The CCs of the elements Hf and Zr are lower than 0.5. Notably, the contents of enriched elements such as V (26.54~886 ppm, avg. 197.4 ppm), Ni (13.75~212 ppm, avg. 85.8 ppm), Ba (280.6~3898 ppm, avg. 1292.8 ppm), Zn (9.6~4942 ppm, avg. 177.3 ppm), and U (0.82~73.1 ppm, avg. 13.7 ppm) are relatively variable and are higher than those of NASC and PAAS (Table S1).

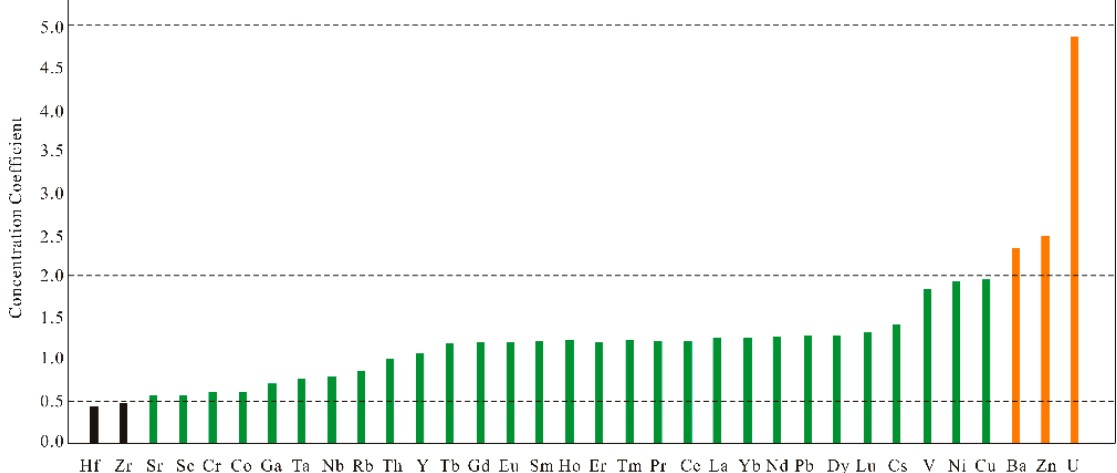

**Figure 10.** Concentration coefficients (CCs) of trace elements in the Wufeng–Longmaxi shale on the Yangtze Platform, South China.

The total rare-earth-element contents ($\sum$REE) in the Wufeng–Longmaxi shale in the Yangtze Platform range from 40.92 to 310.23 ppm, with an average of 184.77 ppm, which is very close to that of the North American shale composition (185.77 ppm) (Table S1). Vertically, $\sum$REE decreases from the interval WF2–3 to WF4 and reaches the minimum in the LM1 interval (Figure 8). Across the LM1 interval, $\sum$REE increases abruptly and reaches its maximum in the LM6 interval. The total light REEs ($\sum$LREE) ranges from 31.3 to 266.07, with an average of 167.16 ppm, accounting for 90.4% of $\sum$REE. The total heavy REEs ($\sum$HREE) ranges from 7.83 to 44.16 ppm, with an average of 17.61 ppm, accounting for 9.6% of $\sum$REE.

The ratio between $\sum$LREE and $\sum$HREE ($\sum$LREE/$\sum$HREE) reflects the fractionation degree of light and heavy REEs. A higher value of $\sum$LREE/$\sum$HREE indicates more enrichment of LREE. The ratio of $\sum$LREE/$\sum$HREE fluctuates between 3.25 and 11.88, with an average of 9.46, which is very close to the North American black shales (10.49). In the chondrite-normalized diagrams, all samples show a similar REE pattern characterized by a relatively high slope of the LREE part and a flat slope of the HREE part, together with slightly negative Eu anomalies (Eu/Eu* = 0.35~0.92; Figure 11). This pattern reflects a moderate enrichment of LREEs, a consistent provenance, and steady tectonic activity. The $La_N/Yb_N$ values (with the NASC standard) vary from 3.22 to 22.81, with an average of 13.55, which shows a slight LREE enrichment. The values of $La_N/Sm_N$ range from 2.67 to 9.08 (avg. 6.88), indicating an inconspicuous fractionation among HREEs as well.

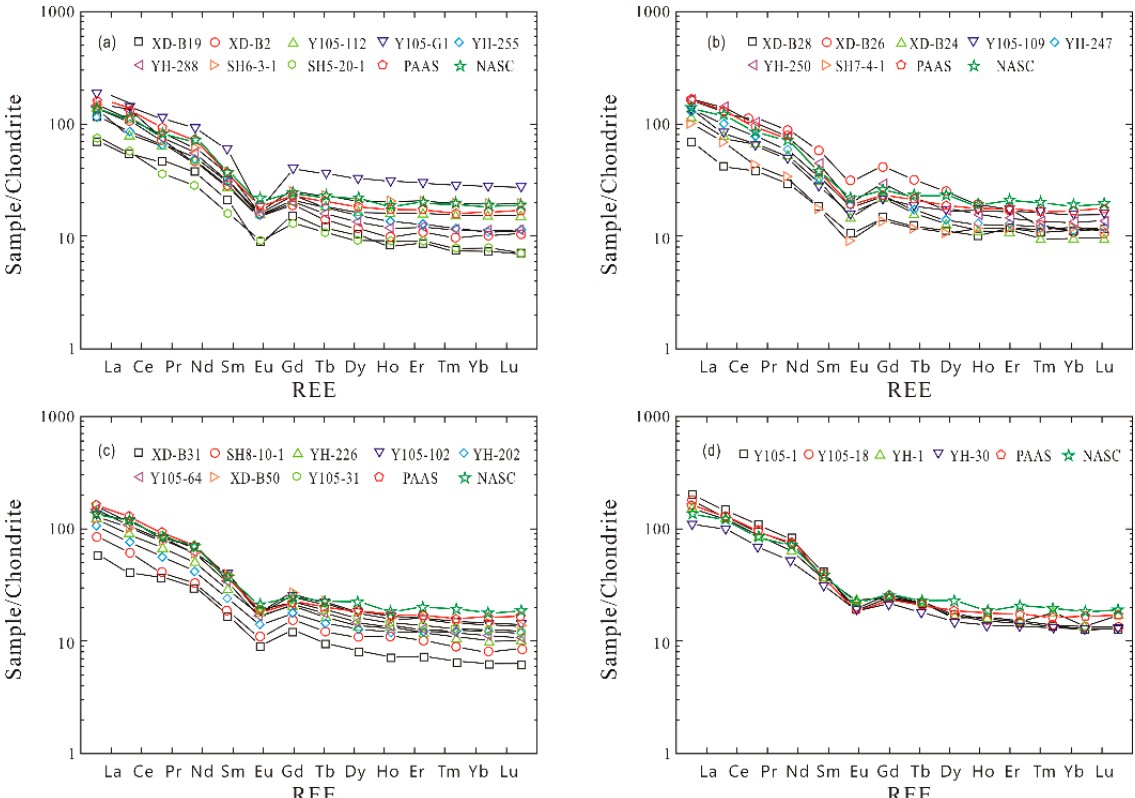

**Figure 11.** Chondrite-normalized rare-earth-element (REE) patterns in sedimentary rocks of the Wufeng–Longmaxi shale in the Yangtze Platform, South China. Chondrites values are from [51]. (**a**) The Wufeng Formation; (**b**) the Guanyingqiao bed; (**c**) the Long 11 member of the Longmaxi Formation; (**d**) the Long 12 of the Longmaxi Formation.

## 5. Discussion

Sedimentary processes (weathering, sorting, and diagenesis) have significant impacts on the compositions of sedimentary rocks and can cause changes in geochemical information [48,52]. Therefore, it is very important to assess the effects of weathering, sorting, and alteration before provenance analysis. Commonly, mobile elements are used to infer the effects of the sedimentary process, whereas immobile elements are utilized to reflect the parent rock lithology and tectonic setting [53,54].

### 5.1. Sedimentary Sorting and Recycling

Hydraulic sorting can result in the enrichment of some heavy minerals (e.g., zircon, monazite, sphene, and apatite) and the depletion of some mobile elements (e.g., Ca, Mg, and Na). Some elements can be used to identify the concentrations of heavy minerals during sediment transport [55,56]. The $TiO_2$–$Al_2O_3$–Zr ternary diagram (Figure 12a), La/Sm versus Th bivariate diagram (Figure 12b), Tb/Yb versus Hf bivariate diagram (Figure 12c), and Ta/La versus Ti bivariate diagram (Figure 12d) are commonly utilized to specify the accumulation of zircon, sphene, and apatite, as well as some allanite. In the $TiO_2$–$Al_2O_3$–Zr ternary diagram (Figure 12a), all samples display a relatively narrow span of $TiO_2$–Zr variations, indicating no or weak sorting of zircon. Furthermore, there are weak and unsystematic correlations observed in these bivariate diagrams (Figure 12b–d), indicating the weak accumulation of sphene, apatite, and zircon [56]. These distributions reveal that sedimentary sorting has had a limited impact on the geochemical features of these samples.

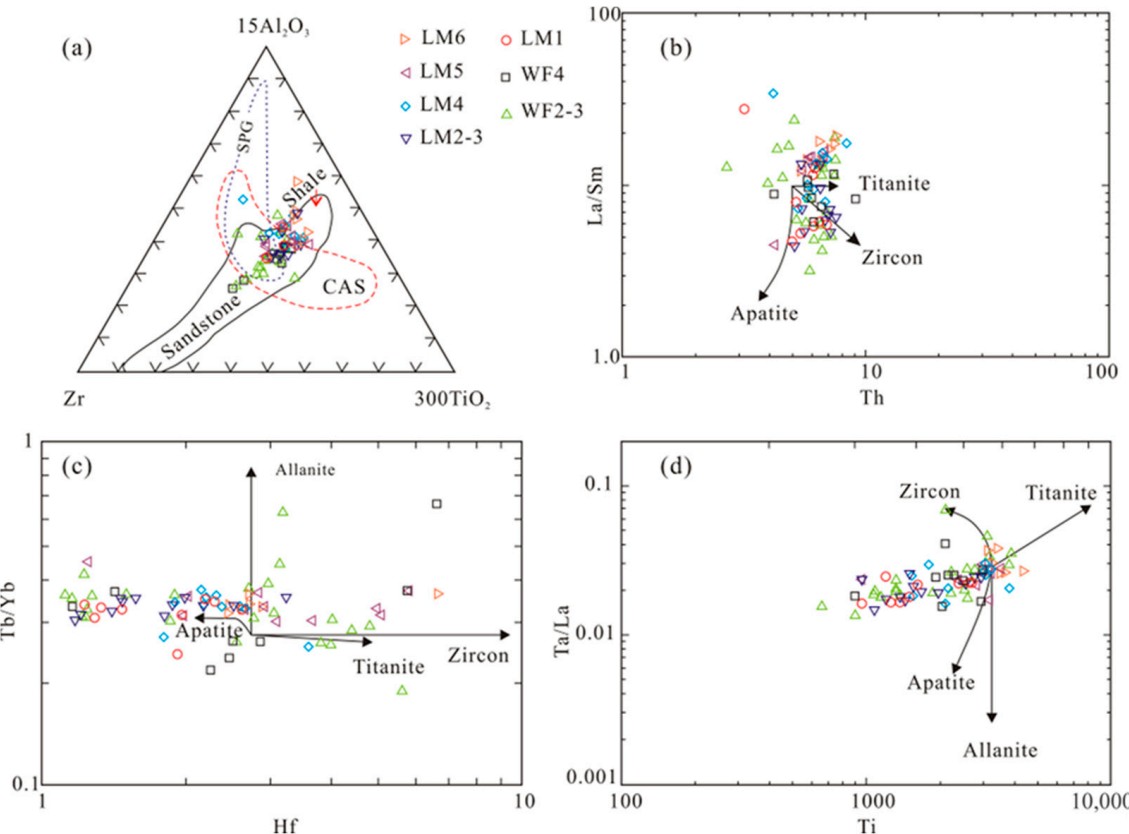

**Figure 12.** (**a**) Al–Ti–Zr triangular diagram [55]; (**b**) La/Sm–Th, (**c**) Tb/Yb–Hf, and (**d**) Ta/La–Ti bivariate diagrams [56] showing the effects of sedimentary sorting. In the figure, CAS refers to the field of calc-alkaline suites, and SPG refers to strongly peraluminous granites.

Zirconium (Zr) commonly exists in zircon, is stable, and can continuously accumulate with sedimentary recycling; scandium (Sc) mainly exists in basic rocks and usually retains the characteristics of sediment source rocks. Therefore, the ratio of Zr/Sc can indicate the enrichment of Zr. Thorium (Th) commonly exists in silicic rocks, and the ratio of Th/Sc can specify the degree of chemical differentiation. As a result, the Th/Sc versus Zr/Sc bivariate diagram can be used to evaluate the effects of sedimentary recycling. The ratios of Th/Sc (0.77–4.36) and Zr/Sc (6.17–43.51) of the studied samples show a high correlation but have no or a slightly increasing trend of Zr/Sc ratios (Figure 13a), indicating no or minor sedimentary recycling of the sediment source rocks.

*5.2. Weathering and Paleoclimate*

The index of compositional variability (ICV), the chemical index of alternation (CIA), and the ratios of Cs/Ti and Rb/Ti can indicate weathering and the paleoclimate. The ICV, which is defined as ICV = (Fe$_2$O$_{3T}$ + K$_2$O + Na$_2$O + CaO* + MgO + MnO + TiO$_2$)/Al$_2$O$_3$, is a common proxy to differentiate the recycling sediment source from the first-cycle one. In detail, samples with ICV < 1 suggest a compositionally mature and recycling sediment source, whereas samples with ICV > 1 suggest a compositionally immature and first-cycle sediment source [61,62]. The CIA, which is defined as CIA = [Al$_2$O$_3$/(Al$_2$O$_3$ + CaO* + Na$_2$O + K$_2$O)] × 100, is a common proxy to quantify the degree of weathering [63]. In general, the CIA value of weakly weathered shale is lower than 65, that of a moderately weathered one is between 65 and 85, and that of an intensely weathered one is between 85 and 100 [64,65]. Weathering is highly correlated with the climate: intense weathering (especially chemical) is related to a warm and humid climate, whereas weak weathering is associated with a cold and arid climate [66]. Here, Fe$_2$O$_{3T}$, K$_2$O, Na$_2$O, CaO*, MgO, MnO, TiO$_2$, and Al$_2$O$_3$ are molecular concentrations, with CaO* representing Ca–silicates only

(i.e., excluding calcite, dolomite, and apatite). In this study, we modified CaO$^*$ based on the ratio of CaO/Na$_2$O [53], and CaO$^*$ equals the small value of mol Na$_2$O and CaO. Cs and Rb are very sensitive to climatic influences and may become leached during chemical weathering [67]. Generally, enhanced chemical weathering will result in higher Cs and Rb contents. Thus, the Cs/Ti and Rb/Ti ratios are important elemental indicators for paleoclimate conditions [67,68].

The ICV values for the studied samples are generally above 1, with an average of 1.69 (Figure 14a), suggesting compositionally immature and first-cycle sediment source rocks. The CIA values for the studied samples vary from 60 to 80, with an average of 68.9 (Figure 14b) after adopting the method of [69], suggesting weak to moderate weathering of the sediment source rocks.

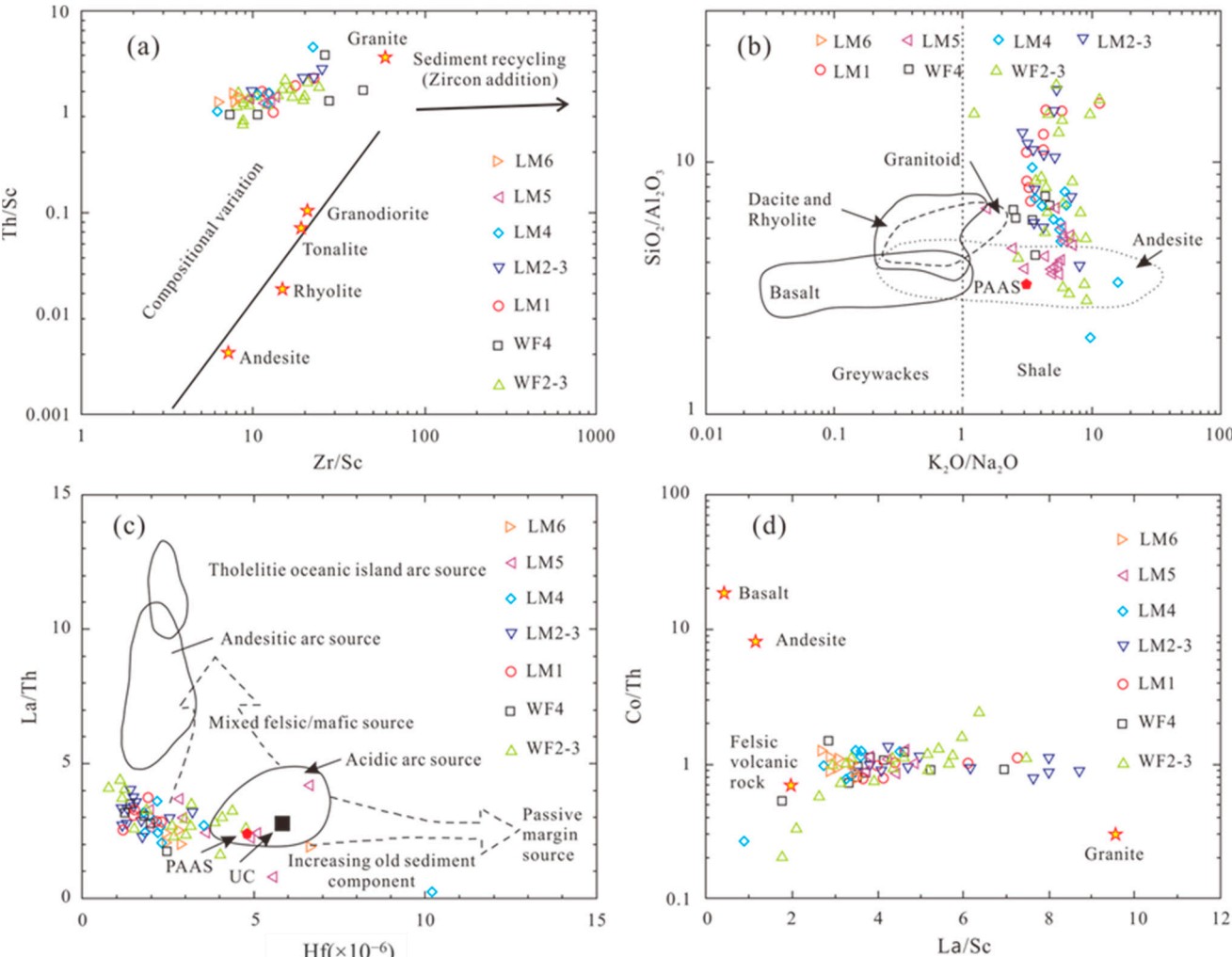

**Figure 13.** (**a**) Th/Sc–Zr/Sc diagram for discriminating sedimentary recycling [53]; (**b**) SiO$_2$/Al$_2$O$_3$– K$_2$O/Na$_2$O [57], (**c**) La/Th–Hf [58], and (**d**) Co/Th–La/Sc diagrams [59] for discriminating source compositions. Compositions of andesite, rhyolite, tonalite, granodiorite, and granite are from [60]. Compositions of PAAS and UC are from [45]. PAAS = post–Archean Australian shales; UC = average upper crust.

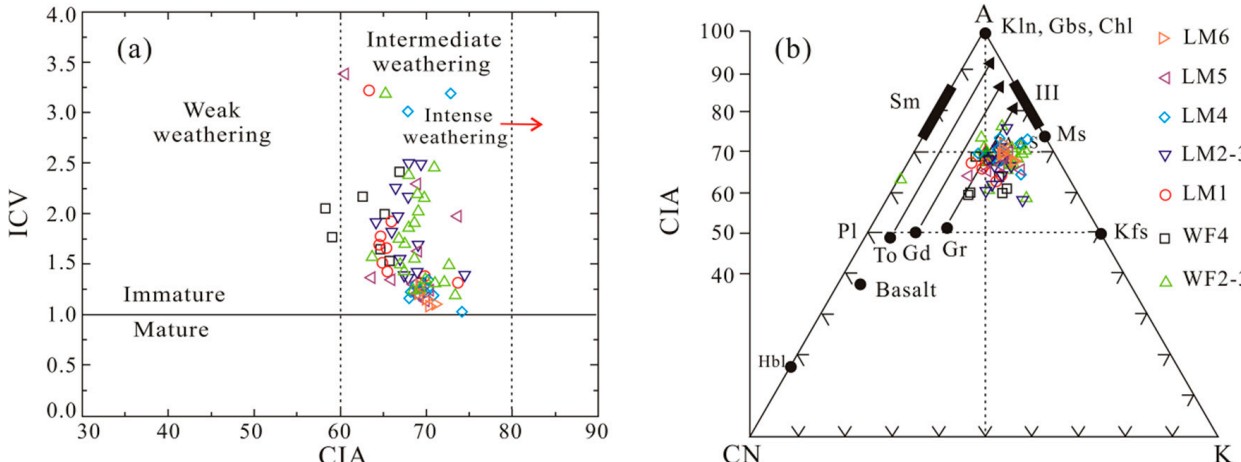

**Figure 14.** (**a**) Index of compositional variability (ICV)–chemical index of alternation (CIA) diagram. (**b**) A–CN–K (Al$_2$O$_3$ − [CaO* + Na$_2$O] − K$_2$O) diagram. Data for tonalite (To), granodiorite (Gd), granite (Gr), and post–Archean Australian shales are from [45]. Arrowheads show the predicted weathering trends. Chl = chlorite; Gbs = gibbsite; Hbl=hornblende; Ill = illite; Kfs = potassic feldspar; Kln = kaolinite; Ms = muscovite; Pl = plagioclase; Sm = smectite.

The values of CIA, Cs/Ti, and Rb/Ti display systematic variations with two obvious shifts (Figure 6). One shift occurs across WF2–3, WF4, and, subsequently, LM1, and the other occurs across LM4 and LM5. For the first shift, WF2–3 have CIA values between 67.5 and 72.7 and relatively higher ratios of Cs/Ti and Rb/Ti (Figure 6), indicating a moderately warm and semi-humid climate and moderate chemical weathering. In contrast, the lower part of WF4 has CIA values between 58.3 and 59.04 and relatively low ratios of Cs/Ti and Rb/Ti, suggesting a cool and arid climate and weak chemical weathering. Subsequently, the values of CIA, Cs/Ti, and Rb/Ti abruptly shift across the uppermost part of WF4, suggesting that the weather changed from cool and arid to moderately warm and semi-humid. The shift is consistent with the two extinction pulses of the LOME [10,15,70,71]. For the second shift, LM4 has CIA values between 69.3 and 70.8, indicating a warm and semi-humid climate and moderate chemical weathering (Figure 6). In comparison, the lowermost part of LM5 has CIA values between 60.5 and 63.8, suggesting a cold and arid climate and weak chemical weathering. This shift is probably related to regional activated tectonism [72].

There is a huge difference in the relative concentrations of REEs between terrigenous sediment and seawater. Generally, the REE contents of terrigenous input are much higher than those of seawater. The mixing of a small amount of terrigenous input may significantly increase the value of REEs in seawater. From WF2–3 to LM1 (Figure 6), the value of ∑REE gradually decreases, indicating the progressive weakening of terrigenous input. From LM1 to LM6, the values of ∑REE gradually increases, indicating the progressive strengthening of terrigenous input.

### 5.3. Protoliths

With respect to protoliths, the sediment source compositions have a close relationship with immobile elements. As a result, the A–CN–K diagram is widely utilized to identify protoliths. All samples are plotted in the granite trend in the A–CN–K diagram (Figure 14b), indicating that the protoliths of these rocks are probably acidic igneous rocks. In addition, the Th/Sc–Zr/Sc bivariate diagram is widely used to identify protoliths. In this diagram, all samples are plotted in and around the area of granite (Figure 13a), further suggesting that the protoliths of these rocks are acidic igneous rocks.

The element content in shale is more homogeneous and stable and can retain most of the information of protoliths [73,74]. As a result, REE patterns can be an effective method for identifying protoliths. Generally, basic rocks have low REE contents and no or positive

Eu anomalies, whereas acidic rocks have high REE contents and obvious negative Eu anomalies [48,75]. Almost all samples display moderate to high REE contents (Table S1) and obvious negative Eu anomalies, indicating that the protoliths could have originated from acidic rocks. In addition, these samples have REE patterns with enriched light REEs and depleted heavy REEs (Figure 11), precluding the contribution of basic rocks.

The ratios between immobile major and trace elements, such as Th/Sc, Zr/Sc, $SiO_2/Al_2O_3$, $K_2O/Na_2O$, La/Th, Co/Th, and La/Sc, can also be used to identify protoliths. In the $SiO_2/Al_2O_3$–$K_2O/Na_2O$, $Zr/Al_2O_3$–$TiO_2/Zr$, La/Th–Hf, and Co/Th–La/Sc binary plots, all samples are plotted between the fields of felsic igneous rock and granite (Figure 13). These further indicate that the protoliths could be felsic igneous rocks.

### 5.4. Tectonic Setting

Commonly, there exists a close relationship between the geochemical compositions of sedimentary rocks and the tectonic setting [53,76,77]. From the oceanic island arc to the continental island arc to the active continental margin to the passive margin, the $TiO_2$, $Al_2O_3$, and $Fe_2O_{3T}$ + MgO concentrations and the $Al_2O_3/SiO_2$ ratios of shale would decrease, while $SiO_2$ concentration and $K_2O/Na_2O$ and $Al_2O_3/(CaO + Na_2O)$ ratios would generally increase [76]. In the $TiO_2$–($Fe_2O_{3T}$ + MgO) diagram, most samples are plotted in and around the continental island arc and active continental margin fields (Figure 15a) [76]. In contrast, most samples are plotted in the passive continental margin and active continental margin fields with a few in the arc field in the ($K_2O$ + $Na_2O$)–$SiO_2$ diagram (Figure 15b).

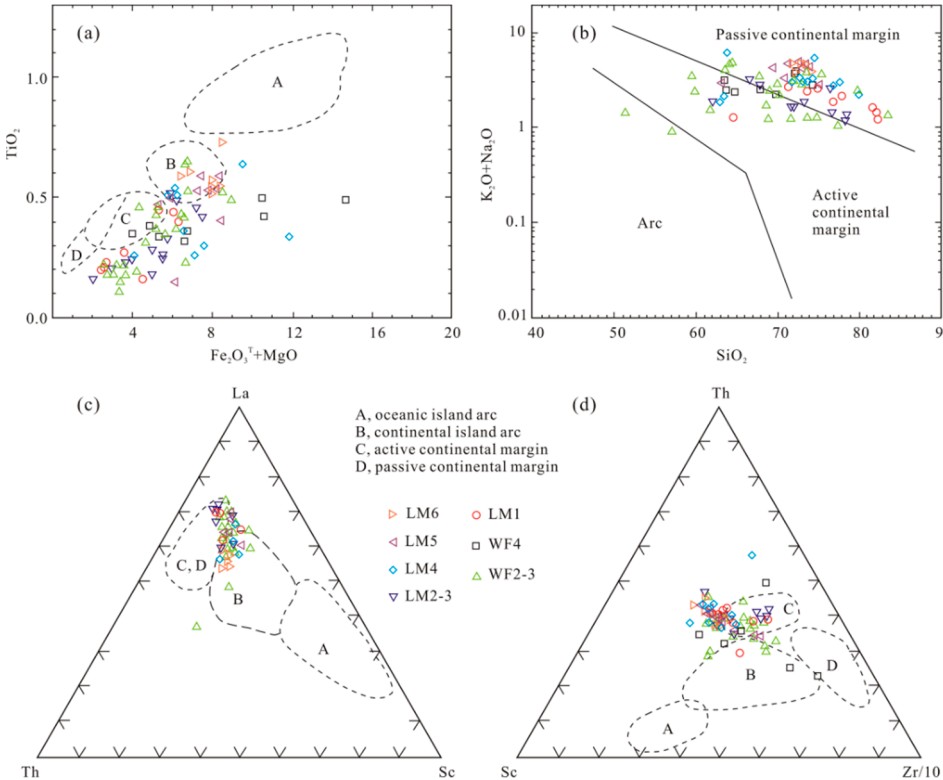

**Figure 15.** Tectonic setting discrimination diagrams for the Wufeng–Longmaxi shale on the Yangtze Platform, South China. (**a**) $TiO_2$–$Fe_2O_{3T}$ + MgO (Bhatia, 1983), (**b**) $K_2O$ + $Na_2O$–$SiO_2$, (**c**) La–Th–Sc, and (**d**) Th–Sc–Zr/10 diagrams [75].

Relatively immobile trace elements such as REEs and HFSEs are also effective in discriminating the tectonic setting of sedimentary rocks. Systematic increases in La, Ce, and HFSEs and in the ratios of Th/U, La/Sc, and Th/Sc have been reported in graywackes from the oceanic island arc to the passive continental margin, together with decreases in Eu/Eu*, Zr/Hf, Zr/Th, La/Th, and Ti/Zr ratios [48,75]. For the Wufeng–Longmaxi shale, most

samples are plotted in and around the active continental margin and continental island arc fields in the Sc–La–Th (Figure 15c) and Zr/10–Th–Sc diagrams (Figure 15d), indicating that the sediment source rocks most likely formed in active continental margin and continental island arc tectonic settings [75].

Major elements such as Na, K, Ca, and Mg are easily weathered and migrate during transportation, diagenesis, and metamorphism [76]. Therefore, the results obtained from major element analysis should be used with caution. However, the results obtained from immobile trace elements are commonly reliable [78,79]. In view of this, we suggest that the tectonic setting of Wufeng–Longmaxi shale is the continental island arc and active continental margin, as inferred from immobile trace elements. This conclusion is consistent with the insight that the Yangtze Plate subducted against the Cathaysian Plate during the formation period of the Wufeng–Longmaxi shale, as acquired in a previous study.

### 5.5. Genesis of Shale Gas Sweet-Spot Interval

The term "sweet spot" for petroleum geology was proposed in [80], which refers mainly to gas-bearing or gas-producing geographic areas with the best enrichment in unconventional shallow biogenic gas basins. This term was subsequently applied to the evaluation of unconventional oil and gas resources [81]. Chinese scholars have widely utilized this term and further expanded its original connotation. The oil and/or gas accumulation intervals in rock formations are named "sweet-spot intervals", while accumulation areas in geography are called "sweet-spot areas" [28].

The shale gas sweet-spot interval of the Wufeng–Longmaxi shale is LM1 (Figure 5). This interval is rich in graptolites such as *Avitograptus avitus* and *Avitograptus ex gr.avitus* [44] and has an $R_o$ ranging from 2.38%~3.02%. The interval is characterized by high TOC content (>2% on average) [82], high quartz content (avg. 60%), high gas content (>6 m$^3$/t), high porosity (>5%) [83], a high ratio of horizontal permeability to vertical permeability, high organic pore content (>61.4%) [29], and high biological quartz content (>85%). In addition, the interval is dominated by paper lamination and graded lamination composed of siltstone and claystone [29], together with abundant microfractures [41].

The genesis of the sweet-spot interval has always been a research hotspot and has various explanations. Some suggest that expanded anoxia induced by a warm and humid climate is the dominant factor for the formation of organic-rich shale [15,22,23,28,84]. Nevertheless, some studies suggest that rapid transgression because of glacial melting may be the dominant factor [85,86], while the lower terrigenous supply induced by the inactive tectonic setting may be the potential reason for the formation of the sweet-spot interval [24].

Here, we propose that the limited terrigenous supply caused by the inactive tectonic setting may be an alternative explanation for the formation of the sweet-spot interval. The reasons could be as follows: First, during the Late Ordovician to Early Silurian, the compression of the Yangtze and Cathaysia blocks in the Upper Yangtze area progressively strengthened [37], leading to an increased sedimentation rate from LM1 to LM6. The focus interval only has a sedimentation rate varying between 1.7 and 7.5 m/Ma [24], which is considerably lower relative to other intervals. The lower sedimentation rate considerably favors the enrichment of organic matter because of the limited dilution of the terrigenous supply [87]. Second, the LM1 interval has the minimum ∑REE and highest TOC content (Table 1; Figure 5), indicating the weakened influence of terrigenous input in this period [48]. In addition, the significant increase in ∑REE accompanied by the decreasing TOC content from the LM2 to LM6 interval can further confirm the above assumption. Third, on account of the warm and humid climate and the rapid rise in sea level during the period of LM1–LM6 [14,23], it is unreasonable to merely ascribe the formation of the sweet-spot interval to a warm and humid climate and a rapid sea-level rise.

## 6. Conclusions

(1) The Wufeng–Longmaxi shale is predominantly composed of quartz, calcite, dolomite, and clay minerals. LM1 is a sweet-spot interval and has the highest contents of total quartz, microcrystalline quartz, and TOC.

(2) The Wufeng–Longmaxi shale was derived from compositionally immature and first-cycle sediment source rocks and underwent weak to moderate weathering and sorting. Sedimentary sorting has a limited impact on the geochemical features of the shale.

(3) The dominant protoliths of the Wufeng–Longmaxi shale were acidic igneous rocks, and the tectonic setting was an active continental margin and continental island arc.

(4) Apart from expanded anoxia and transgression, the limited terrigenous supply caused by the inactive tectonic setting is an alternative cause of the formation of the sweet-spot interval.

**Supplementary Materials:** The following supporting information can be downloaded at: https://www.mdpi.com/article/10.3390/min12101190/s1, Table S1: Trace element data of shale samples from Wufeng Formation-Longmaxi Formation in this study.

**Author Contributions:** Conceptualization, Z.S. and T.Z.; methodology, Z.S. and S.Z.; software, L.D.; validation, T.Z.; formal analysis, T.Z.; investigation, S.Z.; resources, Z.S.; data curation, L.D.; writing—original draft preparation, Z.S.; writing—review and editing, Z.S.; visualization, Z.S. and F.C.; supervision, S.S.; project administration, S.S.; funding acquisition, Z.S. All authors have read and agreed to the published version of the manuscript.

**Funding:** This research was funded by the 14th Five-Year Plan of the Ministry of Science and Technology of PetroChina, grant number 2021DJ1901.

**Institutional Review Board Statement:** All data in this study are subject to confidentiality agreement of the PetroChina. Institutional review board approval of our institute was obtained for this study.

**Data Availability Statement:** Not applicable.

**Acknowledgments:** We appreciate PetroChina Southwest Oil and Gas Field Company for providing the drilling data. We thank Zhangbin Zhou for useful comments and language editing, which have greatly improved the manuscript. Editors and anonymous reviewers are gratefully acknowledged.

**Conflicts of Interest:** The authors declare no conflict of interest.

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
