# Peer review of "Mineralogy and Geochemistry of the Upper Ordovician and Lower Silurian Wufeng-Longmaxi Shale on the Yangtze Platform, South China: Implications for Provenance Analysis and Shale Gas Sweet-Spot Interval"

_minerals, doi:10.3390/min12101190_

Round 1
Reviewer 1 Report
The aim of the MS is suitable for the Minerals journal; however, the authors should re-organize and re-evaluate their data! First of all, the authors should remove all mineralogical data from Section 2, if the reported mineralogical compositions are from the recent study. More importantly, figures 3 and 4 appear to have been taken from a previous study (e.g., Shi et al., 2022. Energies 15,1618) or were not cited in the MS. If the latter one is valid, please cite related figures in the MS. If the former one is valid, please remove these figures from the MS. Secondly, the authors mentioned "biogenic" quartz in the MS; however, the reported quartz grains in figure 5b are not biogenic. These grains are more likely detrital quartz grains or replacement silica (?). A similar case could also be said for "quartz originated from clay mineral transformation" reported in figure 5c. These grains are detrital individual quartz or silica grains within the clay mineral matrix. Therefore, the authors should remove all biogenic quartz from the MS and combine all the quartz in figure 6. Furthermore, the authors mentioned macerals and DOM in the lines between 228 and 236, and in Figure 8. All these maceral identifications are not accurate! Even though some structured macerals (e.g., funginite, telovitrinite, and fusinite) could be defined by using SEM-EDX data, the reported macerals are not correct in Figure 8. Furthermore, liptinite (as reported incorrectly as extinite) cannot be identified using SEM-BSE images. More critically, pre-Silurian samples could not contain any vitrinite or inertinite. Therefore, the authors should provide DOM data from selected samples. For instance, the authors could select relatively high TOC-content Ordovicain samples and graptolite-bearing Silurian samples. Such data could also provide vitrinite and graptolite reflectance data, which also supports their assumption in section 5.5. I added some corrections and suggestions in the revised MS. Overall, I could suggest a major correction and would like to reconsider after the suggested corrections are made.

Author Response
Review 1:
The aim of the MS is suitable for the Minerals journal; however, the authors should re-organize and re-evaluate their data!
[Answer-1-00]
We have revised the MS carefully according to the uploaded file. We have benefited greatly. Here, we want to thank the reviewer very much for your serious and prudent work.
- First of all, the authors should remove all mineralogical data from Section 2, if the reported mineralogical compositions are from the recent study.
[Answer-1-01]
Thanks very much. We have removed all mineralogical data from Section 2 to Section 3.
- More importantly, figures 3 and 4 appear to have been taken from a previous study (e.g., Shi et al., 2022. Energies 15,1618) or were not cited in the MS. If the latter one is valid, please cite related figures in the MS. If the former one is valid, please remove these figures from the MS.
[Answer-1-02]
Thank you very much for your close attention to the author's previous research. The figures are our recent work and are different from our previous study. In this revised manuscript, we cited the references also.
- Secondly, the authors mentioned "biogenic" quartz in the MS; however, the reported quartz grains in figure 5b are not biogenic. These grains are more likely detrital quartz grains or replacement silica (?). A similar case could also be said for "quartz originated from clay mineral transformation" reported in figure 5c. These grains are detrital individual quartz or silica grains within the clay mineral matrix. Therefore, the authors should remove all biogenic quartz from the MS and combine all the quartz in figure 6.
[Answer-1-03]
Thank you very much. We have revised the corresponding parts in the MS.
- Furthermore, the authors mentioned macerals and DOM in the lines between 228 and 236, and in Figure 8. All these maceral identifications are not accurate! Even though some structured macerals (e.g., funginite, telovitrinite, and fusinite) could be defined by using SEM-EDX data, the reported macerals are not correct in Figure 8. Furthermore, liptinite (as reported incorrectly as extinite) cannot be identified using SEM-BSE images. More critically, pre-Silurian samples could not contain any vitrinite or inertinite. Therefore, the authors should provide DOM data from selected samples. For instance, the authors could select relatively high TOC-content Ordovicain samples and graptolite-bearing Silurian samples. Such data could also provide vitrinite and graptolite reflectance data, which also supports their assumption in section 5.5. I added some corrections and suggestions in the revised MS.
[Answer-1-04]
We are very sorry for our mistakes and imprudent study. In the revised MS, we have deleted the corresponding words and figures. In addition, we selected 2 samples with relatively high TOC graptolites contents. With the aid of the thin sections, we recognized organic matters in reflected white light (see Figure 6 in the revised MS).
- Overall, I could suggest a major correction and would like to reconsider after the suggested corrections are made.
[Answer-1-05]
Thanks very much. We hope that the revised MS can meet your requirements.
Reviewer 2 Report
Dear authors ,
Comments are attached

Author Response
Review 2:
The paper is nice and I am happy to read this paper. My only comments:
- Please used lower and upper for rocks, So please change Late Ordovician and Early Silurian Wufeng–Longmaxi Shale to upper Ordovician and lower Silurian Wufeng–Longmaxi Shale.
[Answer-2-01]
Thanks very much. We have revised the corresponding parts in the MS.
- Some proxies are found in the figures and not discussed in the text (e.g., cyclicity, biozones....).
[Answer-2-02]
Thanks very much. We have supplemented the corresponding discussions in the MS.
- The paper needs to compare the Ordovician and Silurian hot shale in the Arabian plate, in addition to organic-rich sediments:
[Answer-2-03]
Thanks very much. We have compared the Wufeng-Longmaxi shale with the Ordovician and Silurian hot shale in the Arabian plate in this revised MS.
- Suggest references:
Arthur, M.A., Sageman, B., 1994. Marine black shales: depositional mechanisms and environments of ancient deposits. Annu. Rev. Earth Planet Sci. 22 (1), 499–551.
Paris, F., Verniers, J., Miller, M. A., Melvin, J., & Wellman, C. H. (2015). Late Ordovician–earliest Silurian chitinozoans from the Qusaiba-1 core hole (North Central Saudi Arabia) and their relation to the Hirnantian glaciation. Review of Palaeobotany and Palynology, 212, 60-84
Geert, K., Afifi, A. M., Al-Hajri, S. I. A., & Droste, H. J. (2001). Paleozoic stratigraphy and hydrocarbon habitat of the Arabian Plate. GeoArabia, 6(3), 407-442.
Yıldız, G. (2022). Late Paleozoic-Early Mesozoic paleotectonics of the northern Arabian Plate (SE Turkey) and its role in the Paleozoic petroleum system. Marine and Petroleum Geology, 137, 105529
Rahmani, A., Naderi, M., & Hosseiny, E. (2022). Shale gas potential of the lower Silurian hot shales in southern Iran and the Arabian Plate: Characterization of organic geochemistry. Petroleum
[Answer-2-03]
Thanks very much. We have learned the references carefully. In addition, we have compared the shales with the Wufeng-Longmaxi shale in the MS.
I hope to see this paper published soon.
Reviewer 3 Report
The paper is nice and I am happy to read this paper. My only comments
Please used lower and upper for rocks, So please change Late Ordovician and Early 3 Silurian Wufeng–Longmaxi Shale to upper Ordovician and lower Silurian Wufeng–Longmaxi Shale
some proxies are found in the figures and not discussed in the text (e.g., cyclicity, biozones....).
The paper needs to compare the Ordovician and Silurian hot shale in the Arabian plate, in addition to organic-rich sediments:
Suggest references:
Arthur, M.A., Sageman, B., 1994. Marine black shales: depositional mechanisms and environments of ancient deposits. Annu. Rev. Earth Planet Sci. 22 (1), 499–551.
Paris, F., Verniers, J., Miller, M. A., Melvin, J., & Wellman, C. H. (2015). Late Ordovician–earliest Silurian chitinozoans from the Qusaiba-1 core hole (North Central Saudi Arabia) and their relation to the Hirnantian glaciation. Review of Palaeobotany and Palynology, 212, 60-84
Geert, K., Afifi, A. M., Al-Hajri, S. I. A., & Droste, H. J. (2001). Paleozoic stratigraphy and hydrocarbon habitat of the Arabian Plate. GeoArabia, 6(3), 407-442.
Yıldız, G. (2022). Late Paleozoic-Early Mesozoic paleotectonics of the northern Arabian Plate (SE Turkey) and its role in the Paleozoic petroleum system. Marine and Petroleum Geology, 137, 105529
Rahmani, A., Naderi, M., & Hosseiny, E. (2022). Shale gas potential of the lower Silurian hot shales in southern Iran and the Arabian Plate: Characterization of organic geochemistry. Petroleum
I hope to see this paper published soon.
Author Response
The authors have attempted to investigate the Mineralogy and geochemical analysis of the Ordovician-Silurian Wufeng and Longmaxi Shale on the Yangtze Platform, South China. The analysis were performed as a case study to identify the provenance and the genesis of shale gas sweet-spot interval, via mineralogy and geochemistry, organic matters of the shale using a combination of experimental facility as an optical microscopy, X–ray diffraction analysis, field emission scanning electron, microscopy X-ray fluorescence spectroscopy, and inductively coupled plasma mass spectrometry. Authors should explore the application and impact of their innovative findings rather the study look like a report to understand about the occurrence of different elemental compositions and have discussed some weathering processes, sorting, dominant rocks.
This needs a lot more effort to write it in well-organized manner. So I recommend this can be accepted after major revision. For example, all the figures need to be clearer and the temperature unit needs to be complete. Also I could not see anything innovative rather routine findings as one prepares a report. Further I have provide a point by point comments and concerns
- As I believe that Wufeng and Longmaxi Shales are two different shale types in china thus this study has been achieved in providing mixed interpretation of two such as elemental composition etc. hence make it clear.
[Answer-3-01]
Thanks very much. We revised the MS carefully and separated the Wufeng Formation from the Longmaxi Formation. In addition, we separated all the samples of the Wufeng Formation from the samples of the Longmaxi Formation in figures 11, 12, 13, and 14.
- Improve the abstract, add results appropriately. The first sentences in the abstract contain very minimum information this need to be rewritten.
[Answer-3-02]
Thanks very much. We have revised the abstract carefully.
- In abstract at the line number 14-16 “To identify the provenance and the genesis of shale gas sweet-spot interval, the mineralogy and geochemistry of the shale were investigated by using an optical microscopy, X–ray diffraction analysis, field emission scanning electron, microscopy X-ray fluorescence spectroscopy, and inductively coupled plasma mass spectrometry”. It is a massive sentence; I would say must break into two and write methods employed in separate sentences.
[Answer-3-03]
Thanks very much. We have revised the sentence carefully and broke it into two separate sentences.
- Abstract provides vague statements as at line number 18 it is written “The shale is mainly composed of quartz…which shale composition you are talking about? It looks like a general statement, be specific. explain?
[Answer-3-04]
Thanks very much. We have revised the sentences carefully.
- Introduction provides very little information describing the subject cited above in title and have not made it a concise. It is recommended that they must tell a good story of why and how this research will be applicable on unconventional reservoirs exploitation and development.
[Answer-3-05]
Thanks very much. We have re-wrote the Introduction carefully. In this revision, we focused on the researches applicable on unconventional reservoirs exploitation and development.
- The geological settings section is unnecessarily lengthy. The authors have spent too much effort on describing the geology of these shales. however; this should be concise clear and short.
[Answer-3-06]
Thanks very much. We have rewrote the geological setting section. In this revision, the section is relatively short, clear, and concise.
- Samples and Methods: this section must be separate from stratigraphic columns , results etc. presently its mix of results and is not appropriately describing the methods and materials , is not describing appropriately experiments. What samples you used provide their photos, dimensions of samples, field and their locations, co-ordinates. This is very vague to me.
[Answer-3-07]
Thanks very much. We have revised this section carefully.
- Core photographs and thin-section photomicrograph showing characteristics of these shales must be provided.
[Answer-3-08]
Thanks very much. We have provided core photos (Figure 3) and thin-section photos (Figure 8) of the shale in the revised MS.
- Sections 5, in particular to sub-sections from 5.2 to 5.4 need to be explained properly.
[Answer-3-09]
Thanks very much. We have revised the corresponding parts carefully.
- The characterization results of unconventional reservoir rocks are less significant, plus the discussion of the results is not adequate. These are main part of the entire manuscript; however, I didn't see any bright spot in these parts.
[Answer-3-10]
Thanks very much. We have revised the corresponding parts carefully.
- Many of the sentences are poor and vague hence you must correct and proof read the manuscript for correction of mistakes.
[Answer-3-11]
Thanks very much. We have revised the MS carefully.
- Figure 10, 11,12,13,14 are poor quality of these must be improved. Figures Captions are unnecessarily explained. You could provide details in the text rather in captions of figures.
[Answer-3-12]
Thanks very much. We have improved the quality of the figures. In addition, the corresponding captions are revised carefully.
- What is the relation between the petro-physical and the kinds of facies etc.
[Answer-3-13]
Thanks very much. We have deleted the corresponding discussions since they are not related to the subject of this article.
- Conclusions are very poor
[Answer-3-14]
Thanks very much. We have revised the Conclusions.
Round 2
Reviewer 2 Report
The paper is well written, presents new information related to shale formation, and can be published with minor revisions.
i. Number of keywords provided in the manuscript is more than required and must be shortened.
ii. spell check is essential and grammar style shoud be checked, in some places sentences unnecessarily lengthy should be shortened.
iii. Tables and figures and their numbers should be revisited
iv. Finally if you could revisit the conclusions
The Paper meets the criteria to be accepted.
Author Response
Please see the detail reply in following attachments.
